# SlimDiff: Training-Free, Activation-Guided Hands-free Slimming of Diffusion Models

## Abstract

Diffusion models (DMs), lauded for their generative performance, are computationally prohibitive due to their billion-scale parameters and iterative denoising dynamics. Existing efficiency techniques, such as quantization, timestep reduction, or pruning, offer savings in compute, memory, or runtime but are strictly bottle-necked by reliance on fine-tuning or retraining to recover performance. In this work, we introduce SlimDiff, an automated activation-informed structural compression framework that reduces both attention and feedforward dimensionalities in DMs, while being entirely gradient-free. SlimDiff reframes DM compression as a spectral approximation task, where activation covariances across denoising timesteps define low-rank subspaces that guide dynamic pruning under a fixed compression budget. This activation-aware formulation mitigates error accumulation across timesteps by applying module-wise decompositions over functional weight groups: query–key interactions, value–output couplings, and feedforward projections — rather than isolated matrix factorizations, while adaptively allocating sparsity across modules to respect the non-uniform geometry of diffusion trajectories. SlimDiff achieves up to $35\%$ acceleration and $\sim 100M$ parameter reduction over baselines, with generation quality on par with uncompressed models without any backpropagation. Crucially, our approach requires only about $500$ calibration samples, over $70\times$ fewer than prior methods. To our knowledge, this is the first closed-form, activation-guided structural compression of DMs that is entirely training-free, providing both theoretical clarity and practical efficiency.

## 1 Introduction

Diffusion models (DMs) (Rombach et al. (2022); Ramesh et al. (2022); Saharia et al. (2022)) have become the dominant paradigm in generative modeling, achieving remarkable performance. Their power, however, comes at a steep computational cost: every sample requires hundreds of denoising iterations, each iteration invoking a billion-parameter U-Net architecture (Dhariwal & Nichol (2021)). The sequential reliance on such high-dimensional operators leads to substantial latency, memory, and energy demands, making real-time or resource-constrained deployment prohibitive.

Structural slimming offers a direct avenue for reducing both parameters and MACs (Shen et al. (2025)), yet applying it to DMs exposes fundamental challenges that prior work has largely overlooked. Attention (Vaswani et al. (2017)), a primary building block of the diffusion U-Net, illustrates this difficulty: compressing weight matrices in isolation neglects the coupled nature of effective computations, which arise from products like query-key interactions ($\mathcal{QK}$), value-output couplings ($\mathcal{VO}$), and feedforward projections ($\mathcal{FFN}$). Since the rank of a product is bounded by the smallest rank among its factors (Kolter (2007)), maximal compression can be attained only when these products are treated as functional units (Lin et al. (2024)). As shown in App A.6, DM weights are largely high-rank with heavy-tailed spectra, and truncation errors accumulate through sequential denoising. Effective compression is achieved when the weight structure aligns with activation correlations, which define the active subspaces during denoising. Thus, optimal compression requires *data awareness* (Lin et al. (2024)). Ignoring these structural dependencies leads to suboptimal compression decisions.

The second issue is that compressibility in DMs is inherently timestep-dependent. Activation correlations reveal a far richer low-rank structure than weights alone, but the their covariance evolves

dramatically over the denoising trajectory (Wang et al. (2024a)). The activation distribution differs not only across timesteps but also across functional modules, each interacting with the weights in distinct ways. Approximating weights with a static, timestep-agnostic basis therefore collapses this evolving geometry and leads to poor preservation of fine details (Yao et al. (2024b)).

A third issue is error propagation. In diffusion, compression errors are not local; distortions introduced at one step are passed through every subsequent denoising update. Small deviations in early layers compound multiplicatively, creating irreversible degradation (Zeng et al. (2025)). Existing methods allocate sparsity myopically, without accounting for this sequential amplification, and therefore, rely on costly fine-tuning or retraining with large datasets to recover lost performance. Such dependence on retraining undermines the very motivation for slimming. (Zhang et al. (2024a))

Finally, data-aware slimming requires collecting activations across timesteps and prompts (Lin et al. (2024)), and exhaustive sampling is computationally infeasible. Since naïve calibration over thousands of prompts is cumbersome, a principled strategy is needed to select a compact yet representative subset of prompts that spans the relevant activation subspace (Nguyen & He (2025)).

Prior works ( Zhang et al. (2024b); Gao et al. (2024)) exemplify these limitations: although they reduce parameter counts, they remain functional module-agnostic and rely on finetuning or distillation on a large dataset to correct for timestep-dependent distortions and error propagation. Other approaches, such as (Lu et al. (2022)) and ( Chen et al. (2025)), design compact models from scratch but at the cost of prohibitively expensive retraining. Alternatively, a parallel line of work orthogonally looks at inference-time accelerations such as (Wang et al. (2024a); Bolya & Hoffman (2023); Fang et al. (2023)) that reduce computation by truncating denoising timesteps or merging tokens at run-time. These methods incur runtime overhead at every invocation and yield savings that fluctuate across runs - making both the effective cost and the achievable compression level unpredictable. Other methods explore quantization (Li et al. (2023); Zeng et al. (2025)) strategies to accelerate inference, which can be used in parallel with our structural slimming method.

In this paper, we introduce SlimDiff, the first training-free, activation-guided framework that addresses structural, temporal, and propagation-aware challenges of DM slimming in a unified platform. Our contributions are:

- **Principled compression design:** We introduce *module-aligned decompositions* that compress functionally related weight groups instead of isolated matrices, ensuring the compressed model remains structurally consistent with the diffusion computation graph.

- **Data- and process-aware compression:** To align compression with the dynamics of denoising, we propose *timestep-aware compressibility*, leveraging activation statistics stratified by timestep, and *propagation-aware rank allocation*, which globally distributes sparsity under an explicit model of error amplification.

- **Efficient calibration:** We design *SlimSet*, a compact semantic-aware calibration set of only 500 prompts–over $70\times$ fewer than prior works–that spans representative compressible subspaces, making the entire activation collection pipeline lightweight and practical.

- **Comprehensive validation:** We evaluate SlimDiff on MS-COCO (Lin et al. (2014)), LAION Aesthetics (Schuhmann et al. (2022)), ImageReward (Xu et al. (2023)), and PartiPrompts (Yu et al. (2023)) across DMs SDv1.5 and SDv1.4 (Rombach & Esser (2022b;a)). SlimDiff reduces $\sim 100M$ parameters, reduces FLOPs by $22\%$, and speeds up inference by $35\%$ while preserving quality. We also confirm robustness via human preference scoring on HPS v2.1( Wu et al. (2023)), ImageReward, and Pic-a-Pic v1( Kirstain et al. (2023)).

## 2 METHODOLOGY

We aim to compress a pretrained stable diffusion model $\Theta$ into a compact model $\hat{\Theta}$ that satisfies a full parameter budget $B$ while preserving output quality. Formally, we pose this as a constrained optimization problem:

$$\min_{\hat{\Theta}} \; \mathcal{L}_{\text{qual}}(\hat{\Theta}) \quad \text{s.t.} \quad \text{params}(\hat{\Theta}) \leq B \tag{1}$$

where $\mathcal{L}_{\text{qual}}(\hat{\Theta}) = \mathbb{E}_{p,t}\left[\|f_{\Theta}(x_t, p) - f_{\hat{\Theta}}(x_t, p)\|^2\right]$ measures the expected reconstruction error between the original and compressed models across prompts $p$ and timesteps $t$. Rather than optimizing this intractable objective directly, SlimDiff decomposes it into per-module surrogate objec-

Figure 1: **SlimDiff** compresses diffusion models by sampling a semantic calibration set (**SlimSet** $\mathcal{S}$), Spectral Influence Scoring each module's alignment with input anisotropy, which drives Timestep-Aware Correlation Modeling and an Automatic Rank Allocator under a global budget. Finally, MADAC applies whitening–SVD to $\mathcal{QK}$/$\mathcal{VO}$ and Nyström reduction to $\mathcal{FFN}$.

tives (Eq. 5) that admit closed-form solutions. These closed-form surrogates operate at the level of *functional weight groups*: self-attention query–key pairs ($\mathcal{QK}$), value–output pairs ($\mathcal{VO}$), and feed-forward blocks ($\mathcal{FFN}$), rather than individual matrices. For each group, SlimDiff uses activation correlations to construct a activation-guided low-rank approximation that minimizes $\mathcal{L}_{\text{qual}}$ locally under a rank constraint. Our pipeline in figure 1 outlines four key components: **(i) Spectral Influence Scoring:** For each module, we quantify alignment with dominant activation directions to measure its relative importance (Sec. 2.1). **(ii) Semantic Calibration Dataset (SlimSet):** From the full prompt pool $\mathcal{P}$, we select a compact subset $\mathcal{S}$ that spans activations across timesteps, using geometric-median clustering with furthest-point sampling (FPS) in embedding space ( Nguyen & He (2025)) (Sec. 2.2). **(iii) Timestep-aware Correlation Modeling:** Using SlimSet activations, we compute per-timestep correlations for $\mathcal{QK}$, $\mathcal{VO}$, and $\mathcal{FFN}$ modules. These are aggregated into fidelity-weighted mixtures, assigning greater weight to timesteps most salient to output quality (Sec. 2.3). **(iv) Module-Aligned Data-Aware Compression (MADAC) and Rank Allocation:** Aggregated correlations drive a modular compression objective with tailored decompositions for $\mathcal{QK}$, $\mathcal{VO}$, and $\mathcal{FFN}$ blocks (Sec. 2.4). Influence scores then guide a *propagation-aware* per-layer rank selection under a user-specified parameter budget (Sec. 2.5), assigning higher ranks to modules whose errors accumulate most strongly over the denoising trajectory. Together, these stages yield a closed-form, training-free pipeline that 'slims' diffusion models without performance degradation, delivering substantial reductions in parameters and FLOPs without retraining.

## 2.1 ANCHORING METRIC: SPECTRAL INFLUENCE SCORE

At the core of SlimDiff is an anchoring metric, we call the *Spectral Influence Score*, which quantifies how strongly each module aligns with the anisotropy of its input activations. This score serves as the foundation for both timestep-aware correlation accumulation and rank allocation, ensuring that compression decisions remain faithful to the evolving geometry of diffusion activations.

Formally, we adopt the trace-normalized Rayleigh quotient (TRQ) ( Chen (2020)) as the spectral influence score. For a weight matrix per layer $l$, $\mathcal{W}_l$ and pre-activation covariance $\widetilde{\mathcal{C}}_{l,t}$ at timestep $t$, we define our influence score $\mathcal{I}_{l,t}$ as:

$$\mathcal{I}_{l,t} = \text{TRQ}_{l,t}(\mathcal{W}_l) = \frac{\text{tr}(\mathcal{W}_l^\top \widetilde{\mathcal{C}}_{l,t} \mathcal{W}_l)}{\|\mathcal{W}_l\|_F^2 \, \text{tr}(\widetilde{\mathcal{C}}_{l,t})} = \frac{\| \widetilde{\mathcal{C}}_{l,t}^{1/2} \mathcal{W}_l \|_F^2}{\|\mathcal{W}_l\|_F^2 \, \text{tr}(\widetilde{\mathcal{C}}_{l,t})} \tag{2}$$

This formulation is variance-invariant: normalizing by $\text{tr}(\widetilde{\mathcal{C}}_{l,t})$ cancels stepwise scaling, and scale-invariant, as normalizing by $|\mathcal{W}_l|_F^2$ removes dependence on parameter norms. Importantly, it isolates directional alignment: in the eigenbasis of $\widetilde{\mathcal{C}}_{l,t}$, the score evaluates how strongly $\mathcal{W}_l$ projects onto high-variance directions, focusing on anisotropy rather than raw energy. Aggregating TRQ across timesteps with a convex mixture (Sec. 2.3) yields a single spectral influence score per module $\mathcal{I}_l$, which we use to construct a single data-aware correlation per module and to anchor the overall compression objective. These scores drive a *propagation-aware* rank allocation, assigning higher ranks to modules whose anisotropic directions contribute most to error accumulation along the denoising trajectory.

## 2.2 SLIMSET: SEMANTIC-AWARE CALIBRATION DATASET FORMATION

Activation-guided compression requires estimating correlations $\Sigma_{l,t} = \mathbb{E}[X_{l,t}^{\top} X_{l,t}]$ across timesteps and modules. However, collecting these statistics over the full prompt corpus $\mathcal{P}$ is computationally expensive. We therefore introduce **SlimSet**, a semantic coreset $\mathcal{S}$ that preserves the statistical geometry of $\mathcal{P}$ while reducing calibration cost by more than $70\times$. Note that, SlimSet construction is lightweight, taking only a few minutes, and follows the semantic coreset selection strategy introduced in SCDP ( Nguyen & He (2025)).

**Semantic embedding.** Each prompt $p_i \in \mathcal{P}$ is embedded into $E_i \in \mathbb{R}^d$ using CLIP, providing a semantic space where distances reflect prompt similarity. We compute the geometric median $c$ of the embedding cloud, and assign each prompt a distinctiveness score $f_i = \|E_i - c\|_2$ (also denoted as *FD*). Large $f_i$ (far from median $c$) captures rare semantics, small $f_i$ (near $c$) captures common semantics.

**Bin allocation and sampling.** To balance frequent and rare concepts, we stratify prompts into $B$ quantile bins of $\{f_i\}$ and assign each bin a quota $q_b$ proportional to corpus size. Within each bin, we apply farthest-point sampling (FPS):

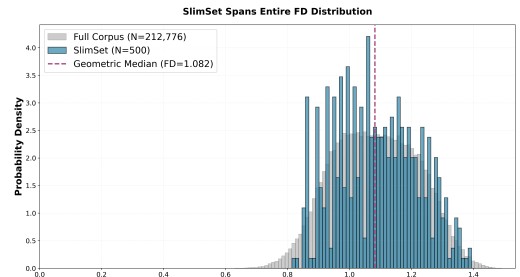

**SlimSet Spans Entire FD Distribution**

$$\mathcal{S}_b = \arg \max_{|\mathcal{S}_b|=q_b} \min_{i \neq j \in \mathcal{S}_b} \big(1 - \cos(E_i, E_j)\big),$$

, ensuring selected prompts are well-spread and mutually diverse. We then perform cosine-based de-duplication to remove redundancy. This two-stage strategy: quantile stratification followed by within-bin FPS, guarantees coverage of both central (low-$f$) and tail (high-$f$) semantics.

**Resulting calibration set.** The final SlimSet $\mathcal{S}$ retains only $J \ll |\mathcal{P}|$ prompts ($J = 500$ vs. 35k), yet yields activation covariances satisfying

Figure 2: **SlimSet coverage.** Distribution of distinctiveness scores for LAION-212K (grey) and SlimSet with $J{=}500$ (blue). Quantile-based binning with proportional allocation ensures SlimSet spans the entire corpus range.

$\Sigma Cov_{l,t}^{\mathcal{S}} \approx \Sigma Cov_{l,t}^{\mathcal{P}}$ across layers $l$ and timesteps $t$. This compactness permits accurate estimation of module-level covariance structure at a fraction of the sampling cost, while preserving both semantic diversity and statistical fidelity. Fig. 3 and Sec. 4(i) detail the SlimSet size ablation and stability evaluation. Appx.7 provides per-dataset coverage analysis and qualitative examples from all quantile bins.

## 2.3 TIMESTEP-AWARE CORRELATION MODELING

DM activations evolve over the denoising trajectory, with correlation structure strongly tied to timestep. A single, timestep-agnostic covariance thus misrepresents the statistics: early steps are near-isotropic noise, while later ones show anisotropic, perceptually aligned variance (App. Sec A.5, Fig. 5). To capture this, we compute *timestep-aware correlations* per module and form a fidelity-weighted mixture emphasizing steps relevant to output quality.

Let $x_{l,t}^{(i)} \in \mathbb{R}^d$ denote the $i$-th activation sample (corresponding to the $i$-th prompt) at layer $l$ and timestep $t$. Let $\mathcal{X}_{l,t} \in \mathbb{R}^{N_t \times d}$ denote the data matrix stacking all $N_t$ spatial samples across prompts. We compute the empirical second moment as:

$$\widehat{\mathcal{C}}_{l,t} = \tfrac{1}{N_t} \mathcal{X}_{l,t}^{\top} \mathcal{X}_{l,t} = \tfrac{1}{N_t} \sum_{i=1}^{N_t} x_{l,t}^{(i)} (x_{l,t}^{(i)})^{\top} \in \mathbb{R}^{d \times d} \tag{3}$$

To ensure numerical stability and invertibility (especially with limited prompts), we use a regularized estimate: $\widetilde{\mathcal{C}}_{l,t} = \widehat{\mathcal{C}}_{l,t} + \epsilon \mathbf{I}_d$, $\epsilon = 10^{-6}$. This regularized correlation $\widetilde{\mathcal{C}}_{l,t}$ is used in the spectral influence score (Eq. 2). Finally, we aggregate across timesteps using convex weights $w_{l,t} \geq 0$, $\sum_t w_{l,t} = 1$, derived from the spectral influence scores :

$$\bar{\mathcal{C}}_l = \sum_{t=1}^{T} w_{l,t} \widetilde{\mathcal{C}}_{l,t}, \quad \bar{\mathcal{R}}_l = \bar{\mathcal{C}}_l^{1/2} \tag{4}$$

For cross-attention, the text features are time-invariant, so their correlation statistics are computed once over SlimSet and reused.

**Per-module variants.** We apply spectral influence score $\mathcal{I}_{l,t}$ consistently across functional groups: (i) **Self-attention (SA):** $\mathcal{QK}$ logits use $\bar{\mathcal{R}}^{\text{sa}}$; $\mathcal{VO}$ maps use $\bar{\mathcal{R}}^{\text{sa}}$. (ii) **Cross-attention (CA):** Queries use step-mixtures $\bar{\mathcal{R}}^q$, while keys and values use cached text statistics $\mathcal{R}^{\text{text}}$. (iii) **Feed-forward (FFN):** Down-projection $\mathcal{W}_d$ is scored with FFN intermediate correlations $\bar{\mathcal{K}}^{\text{ffn}}$. We show the variation of $\mathcal{I}_{l,t}$ across layers, timesteps and functional modules in App. Sec. A.5, Fig 4.

## 2.4 MODULE-ALIGNED DATA-AWARE COMPRESSION (MADAC)

A core challenge in DM compression is that conventional per-matrix factorization treats weights in isolation, overlooking the functional coupling across attention and feed-forward modules. MADAC addresses this by treating each module as an integrated unit and learning a joint, data-aware decomposition that preserves the end-to-end mapping under empirically observed activation distributions. Building on the principles of Lin et al. (2024), we derive diffusion-specific formulations that adapt joint low-rank decomposition to the unique activation geometry of diffusion models.

In attention, the query–key interaction $\mathcal{X}\mathcal{W}_q\mathcal{W}_k^\top\mathcal{X}^\top$ and the value–output map $\mathcal{X}\mathcal{W}_v\mathcal{W}_o$ are *bilinear* in the weights, so compression must respect cross-matrix coupling rather than factorizing each weight in isolation. Likewise, feed-forward blocks compute $\mathcal{Z} = (\mathcal{X}\mathcal{W}_x) \odot \sigma(\mathcal{X}\mathcal{W}_g)$ and $\mathcal{Y} = \mathcal{Z}\mathcal{W}_D$, where the up-projection $\mathcal{W}_U$ is split into a *content* branch $\mathcal{W}_x$ and a *gate* branch $\mathcal{W}_g$, with down-projection $\mathcal{W}_D$. The elementwise gating in FFN breaks linearity, making per-matrix factorization inadequate. Denoting module input activations as $\mathcal{X}$ and attention projections as $\{\mathcal{W}_q, \mathcal{W}_k, \mathcal{W}_v, \mathcal{W}_o\}$, we represent each functional module as $f(\mathcal{X}; \mathcal{W}_1, \mathcal{W}_2)$ and compress weight groups by minimizing the data-driven reconstruction loss:

$$\min_{\widehat{\mathcal{W}}_1, \widehat{\mathcal{W}}_2} \sum_{i=1}^{N} \left\| f(\mathcal{X}_i; \mathcal{W}_1, \mathcal{W}_2) - f(\mathcal{X}_i; \widehat{\mathcal{W}}_1, \widehat{\mathcal{W}}_2) \right\|_F^2 \tag{5}$$

We solve this optimization using the timestep-aware correlation mixtures $\bar{\mathcal{C}}_l$ from Section 2.3, ensuring compression aligns with the activation geometry encountered during inference. For each functional group, we derive specialized decompositions that respect both the computational structure and activation statistics: Nyström approximation for FFN modules and whitened SVD for attention (QK, VO) modules. They are detailed as follows:

**Type-I: $\mathcal{FFN}$ via Nyström approximation.** The feedforward module consists of gated up-projections $\mathcal{W}_x, \mathcal{W}_g \in \mathbb{R}^{d \times 4d}$ followed by a down-projection $\mathcal{W}_D \in \mathbb{R}^{4d \times d}$. Since $\mathcal{W}_g$ resides within the nonlinearity $\sigma(\cdot)$, we constrain $\mathcal{W}_g, \mathcal{W}_k$'s compressed forms to share a column selection matrix $\mathrm{M}_k \in \mathbb{R}^{4d \times k}$ for tractable optimization of Eq. 5: $\mathcal{W}'_x = \mathcal{W}_x\mathrm{M}_k, \mathcal{W}'_g = \mathcal{W}_g\mathrm{M}_k$. For $\mathcal{W}_D$, we ensure dimensional compatibility with compressed up-projections by searching over $\mathbb{R}^{k \times d}$. Our theoretical analysis (App. Sec. A.1) reveals that when a single column selection matrix is used, Eq. 5 reduces to a Nyström approximation of the intermediate activation correlation matrix.

**Theorem 1** Let $\widehat{\mathcal{W}}_x, \widehat{\mathcal{W}}_g$ be constrained to the form $\mathcal{W}_x\mathrm{M}_k, \mathcal{W}_g\mathrm{M}_k$ where $\mathrm{M}_k$ is a $k$-column selection matrix, and let $\widehat{\mathcal{W}}_D$ be searched over $\mathbb{R}^{k \times d}$. The optimal $\widehat{\mathcal{W}}_D^*$ is given by:

$$\widehat{\mathcal{W}}_D^* = (\mathrm{M}_k^\top\mathcal{K}\mathrm{M}_k)^\dagger\mathrm{M}_k^\top\mathcal{K}\mathcal{W}_D \tag{6}$$

where $\mathcal{K} = \sum_{i=1}^{N} \mathcal{Z}_i^\top\mathcal{Z}_i$ is the intermediate activation correlation matrix and $\mathcal{Z}_i = (\mathcal{X}_i\mathcal{W}_x) \odot \sigma(\mathcal{X}_i\mathcal{W}_g)$. The Type-I reconstruction error satisfies:

$$\mathcal{V}_I \le \|\mathcal{W}_D\|_2^2\|\mathcal{K}^{-1}\|_2 E_{\text{Nys}}(\mathcal{K}) \tag{7}$$

where $E_{\text{Nys}}(\mathcal{K})$ denotes the Nyström approximation error of $\mathcal{K}$ using the same $\mathrm{M}_k$. Theorem 1 shows that effective Type-I compression can be achieved through a well-designed Nyström approximation of the intermediate correlation matrix $\mathcal{K}$. We implement this via Algorithm 1, which normalizes $\mathcal{K}$ to correlation form, computes a randomized dominant basis, and selects informative columns using column-pivoted QR (CPQR). The optimal down-projection $\mathcal{W}'_D$ is then solved in closed form on the selected subspace, ensuring both up-projections respect the gated nonlinearity while adapting to the compressed intermediate space.

---

**Algorithm 1** Type-I $\mathcal{FFN}$ compression via Nyström approximation

---

**Require:** $\mathcal{W}_x, \mathcal{W}_g \in \mathbb{R}^{d \times d_{\text{int}}}$, $\mathcal{W}_D \in \mathbb{R}^{d_{\text{int}} \times d}$, intermediate activations $\mathcal{Z}_i = (\mathcal{X}_i \mathcal{W}_x) \odot \sigma(\mathcal{X}_i \mathcal{W}_g)$,
     correlation $\mathcal{K} = \sum_{i=1}^{N} \mathcal{Z}_i^\top \mathcal{Z}_i$, target rank $k = \lceil (1 - \text{sparsity}) d_{\text{int}} \rceil$
1: $(Q, R, \text{pivot\_idx}) \leftarrow \text{CPQR}(\mathcal{K})$     ▷ *Column-pivoted QR;* `pivot_idx` *is the column order*
2: $\text{M}_k \leftarrow I_{d_{\text{int}}}[:, \text{pivot\_idx}[1{:}k]]$                     ▷ *Select the first k pivot columns*
3: **return** $(\widehat{\mathcal{W}}_x, \widehat{\mathcal{W}}_g, \widehat{\mathcal{W}}_D) \leftarrow (\mathcal{W}_x \text{M}_k, \mathcal{W}_g \text{M}_k, (\text{M}_k^\top \mathcal{K} \text{M}_k)^\dagger \text{M}_k^\top \mathcal{K} \mathcal{W}_D))$ ▷ *Nyström-approximated branches and closed-form down-projection*

---

**Type-II:** $\mathcal{QK}$ **via whitening SVD (WSVD).** We now focus on the query-key interactions within multi-head attention mechanisms. The $\mathcal{QK}$ computation corresponds to the bilinear form $f(\mathcal{X}; \mathcal{W}_q, \mathcal{W}_k) = (\mathcal{X}\mathcal{W}_q)(\mathcal{W}_k^\top \mathcal{X}^\top)$, which depends on how query and key directions align with the input distribution. We compress this bilinear operation by factorizing the query-key cross-product $\mathcal{W}_q \mathcal{W}_k^\top$. To ensure the compression respects the anisotropy of the input distribution rather than treating all directions equally, we first whiten both query and key matrices using their respective timestep-aware covariance roots $\bar{\mathcal{R}}_q, \bar{\mathcal{R}}_k$ (Sec. 2.3): $\widetilde{\mathcal{W}}_q = \bar{\mathcal{R}}_q \mathcal{W}_q, \widetilde{\mathcal{W}}_k = \bar{\mathcal{R}}_k \mathcal{W}_k$.

**Theorem 2** (*Query-Key compression by whitening SVD*). Let $\widehat{\mathcal{W}}_q, \widehat{\mathcal{W}}_k$ be the rank-$r$ compressed matrices obtained by applying SVD to the whitened cross-product $\widetilde{\mathcal{W}}_q \widetilde{\mathcal{W}}_k^\top = \mathcal{U}\Sigma\mathcal{V}^\top$ and unwhitening: $\widehat{\mathcal{W}}_q = \bar{\mathcal{R}}_q^{-1} \mathcal{U}_r, \widehat{\mathcal{W}}_k = \bar{\mathcal{R}}_k^{-1} \mathcal{V}_r \Sigma_r$. Then the Type-II reconstruction error in Eq. 5 satisfies:

$$\mathcal{V}_{II} \leq \sum_{i=r+1}^{\min(d_q, d_k)} \sigma_i^2(\widetilde{\mathcal{W}}_q \widetilde{\mathcal{W}}_k^\top) \tag{8}$$

where $\sigma_i$ are the singular values of the whitened cross-product in descending order. This theorem shows that whitening SVD provides the optimal rank-$r$ approximation of the $\mathcal{QK}$ bilinear operator under the covariance-normalized Frobenius norm, and preserves the most important interaction patterns between queries and keys. The procedure is applied independently to each attention head as detailed in Algorithm 2. The full theoretical analysis is provided in App. Sec. A.1.

**Type-III:** $\mathcal{VO}$ **via Whitening SVD (WSVD).** Finally, we focus on the Type-III module, which involves the value-output matrices. The module has is expressed as: $f(\mathcal{X}) = \mathcal{X}\mathcal{W}_v \mathcal{W}_o$, so we seek general low-rank matrices for compression: $\widehat{\mathcal{W}}_v \in \mathbb{R}^{d_h \times k}, \widehat{\mathcal{W}}_o \in \mathbb{R}^{k \times d_h}$ such that $\widehat{\mathcal{W}}_v \widehat{\mathcal{W}}_o \approx \mathcal{W}_v \mathcal{W}_o$. The subsequent theorem reveals that the reconstruction can be solved optimally by applying SVD to the whitened composite transformation.

**Theorem 3** (*Value-Output compression by whitening SVD*). If we search $\widehat{\mathcal{W}}_v$ and $\widehat{\mathcal{W}}_o$ over $\mathbb{R}^{d_h \times k}$ and $\mathbb{R}^{k \times d_h}$, respectively, the optimum in Eq. 5 is $\widehat{\mathcal{W}}_v = \mathcal{C}^{-1/2} \mathcal{U}_k$ and $\widehat{\mathcal{W}}_o = \Sigma_k \mathcal{V}_k^\top$. Here, $\mathcal{U}\Sigma\mathcal{V}^\top$ and $\mathcal{C} = \sum_{i=1}^{N} \mathcal{X}_i^\top \mathcal{X}_i$ are the SVD of $\mathcal{C}^{1/2} \mathcal{W}_v \mathcal{W}_o$ and input correlation, respectively. The corresponding Type-III reconstruction error in Eq. 5 is exactly the SVD approximation error relative to $\mathcal{C}^{1/2} \mathcal{W}_v \mathcal{W}_o$:

$$\mathcal{V}_{III} = E_{\text{SVD}}^2(\mathcal{C}^{1/2} \mathcal{W}_v \mathcal{W}_o) \tag{9}$$

In practice (Alg. 3), we use the timestep-aware value correlation $\bar{C}_v$ (SA: $\bar{C}^{\text{sa}}$; CA: cached text), compute the SVD of $\bar{C}_v^{1/2} W_v W_o$, keep rank $r$, and unwhiten; we apply this per head and concatenate. This yields the optimal rank-$r$ approximation under the covariance-weighted Frobenius norm, aligning compression with the anisotropy of value activations (Details in App. Sec. A.1).

## 2.5 Automatic Rank Allocation Engine

Given a parameter budget $B$, we choose per–block ranks $\{r_\ell\}$ to maximize fidelity with total parameters $\leq B$. Since estimating block-wise utility is infeasible, we use the *spectral influence* $\mathcal{I}_{\ell,t}$ (Sec. 2.3) as a surrogate and allocate capacity $\propto \sum_t \mathcal{I}_{\ell,t}$, so higher-influence blocks (layerwise and across denoising trajectory) retain more rank. We first form a timestep mixture $\bar{\mathcal{J}}_\ell \simeq \sum_t \alpha_t \mathcal{I}_{\ell,t}$ where $\{\alpha_t\}$ are weights that emphasize early denoising steps, reflecting the fact that errors introduced early propagate and amplify downstream. Thus, $\bar{\mathcal{J}}_\ell$ is both *module-aware* and *propagation-aware*. More details are in App. Sec. A.2.

---

**Algorithm 2** Type-II $\mathcal{QK}$ compression via whitening SVD

---

**Require:** head-specific QK matrices: $\{W_{q,j} \in \mathbb{R}^{d_q \times d_h}, W_{k,j} \in \mathbb{R}^{d_k \times d_h}\}_{j=1}^H$, correlations $\{C_q, C_k\}$, target rank $r = \lceil (1 - \text{sparsity})\, d_{\text{int}}/H \rceil$

1: $R_q, \leftarrow \mathcal{C}_q^{1/2}$; $R_k \leftarrow \mathcal{C}_k^{1/2}$      ▷ *Compute whitening transforms*

2: **for** $j = 1, \ldots, H$ **do**

3:     $\widetilde{\mathcal{W}}_{q,j} \leftarrow R_q \mathcal{W}_{q,j}$; $\widetilde{\mathcal{W}}_{k,j} \leftarrow R_k \mathcal{W}_{k,j}$, $\mathcal{T} \leftarrow \widetilde{\mathcal{W}}_{q,j} \widetilde{\mathcal{W}}_{k,j}^\top$ ▷ *Whiten Q,K; get whitened composite*

4:     $(\mathcal{U}, \Sigma, \mathcal{V}) \leftarrow \text{SVD}(\mathcal{T})$; truncate $\mathcal{U}_r, \Sigma_r, \mathcal{V}_r$      ▷ *SVD and rank truncation*

5:     $\mathcal{W}_{q,j} \leftarrow R_q^{-1} \mathcal{U}_r$, $\mathcal{W}_{k,j} \leftarrow R_k^{-1} \mathcal{V}_r \Sigma_r$      ▷ *Unwhiten compressed matrices*

6: **end for**

7: **return** $(W_q, W_k) \leftarrow \big([W_{q,1}, \ldots, W_{q,H}], [W_{k,1}, \ldots, W_{k,H}]\big)$      ▷ *Concatenate the heads*

---

**Algorithm 3** Type-III $\mathcal{VO}$ compression via whitening SVD

---

**Require:** head-specific VO matrices: $\{W_{v,j} \in \mathbb{R}^{d_v \times d_h}, W_{o,j} \in \mathbb{R}^{d_h \times d_q}\}_{j=1}^H$, value correlation $C_v$, target rank $r = \lceil (1 - \text{sparsity})\, d_{int}/H \rceil$

1: $R_v \leftarrow \mathcal{C}_v^{1/2}$      ▷ *Compute whitening transform*

2: **for** $j = 1, \ldots, H$ **do**

3:     $\widetilde{\mathcal{W}}_{v,j} \leftarrow R_v \mathcal{W}_{v,j}$, $\mathcal{T} \leftarrow \widetilde{\mathcal{W}}_{v,j} \mathcal{W}_{o,j}$      ▷ *Whiten value matrix, get whitened composite*

4:     $(\mathcal{U}, \Sigma, \mathcal{V}) \leftarrow \text{SVD}(\mathcal{T})$; truncate $\mathcal{U}_r, \Sigma_r, \mathcal{V}_r$      ▷ *SVD and rank truncation*

5:     $\mathcal{W}_{v,j} \leftarrow R_v^{-1} \mathcal{U}_r$, $\mathcal{W}_{o,j} \leftarrow \Sigma_r \mathcal{V}_r^\top$      ▷ *Unwhiten*

6: **end for**

7: **return** $(W_v, W_o) \leftarrow \big([W_{v,1}, \ldots, W_{v,H}], [W_{o,1}; \ldots; W_{o,H}]\big)$      ▷ *Concat across heads*

---

**Softmax-style Allocation.** We convert influence scores into retention fractions via a temperature-controlled softmax (App. A.2), which normalizes importance across layers and concentrates capacity on influential blocks while keeping allocations smooth. Intuitively, each block's retained rank depends not only on its own importance but also on its relative standing among all blocks and timesteps, yielding a propagation-aware allocation under the global budget.

**Mapping to Ranks.** Each block's retention fraction is multiplied by its effective width ($d$ for $\mathcal{QK}/\mathcal{VO}$ per head, $4d$ for $\mathcal{FFN}$ intermediates), rounded to hardware-friendly multiples (of 8), and clipped by a minimum rank for stability. The global average sparsity is then adjusted by a simple bisection search to ensure the final parameter count exactly meets the budget $B$. This convex allocation distributes sparsity in a propagation-aware manner: high-influence blocks retain more rank, while less influential ones are slimmed, all under a unified parameter budget.

This allocation engine is convex, closed-form, and propagation-aware: blocks with high spectral influence automatically keep more capacity, while less critical ones are slimmed more aggressively. All mathematical details are provided in the App. Sec. A.2.

# 3 EVALUATION AND ANALYSES

We evaluate SlimDiff on SDv1.4 and SDv1.5, comparing against both uncompressed models and competitive compression baselines. Our study is structured around key research questions, detailed in the following subsections, while the full experimental setup is deferred to the App. Sec. A.4.

**3.1 Does SlimDiff preserve generation quality under compression?** To evaluate SlimDiff's ability to maintain generation quality while achieving significant compression, we conduct comprehensive experiments on the MS-COCO 2014 validation dataset ( Lin et al. (2014)). We benchmark against state-of-the-art diffusion compression methods (BK-SDM, Small Stable Diffusion, LD-Pruner) and also report autoregressive baselines (DALL-E, CogView).

Table 1 presents quantitative results on generation quality. SlimDiff achieves competitive performance with only minimal degradation: FID of 13.12 (vs. 13.07 for SD v1.5) and CLIP score of 0.319 (vs. 0.322 for SD v1.5). Notably, SlimDiff is the only method that achieves this performance while being completely backpropagation-free (BP-free), requiring only 500 data points and 4 A100 days compared to 6250 days for the original model. This represents a significant practical advan-

Table 1: Comparison on MS-COCO ($512 \times 512$, 50 denoising steps, CFG=8). Lower FID and higher IS/CLIP is better. 'BP-free' indicates training-free compression, methods using BP are grayed out. LD-Pruner$^\star$ is not open-sourced; results reported from its paper may not be directly comparable.

| Model | BP-free | # Params | FID↓ | IS↑ | CLIP↑ | Data Size (M) | A100 Days |
|---|---|---|---|---|---|---|---|
| SD v1.5 (Rombach & Esser (2022b)) | – | 1.04B | 13.07 | 33.49 | 0.322 | > 2000 | 6250 |
| SD v1.4 (Rombach & Esser (2022a)) | – | 1.04B | 13.05 | 36.76 | 0.296 | > 2000 | 6250 |
| Small Stable Diffusion (OFA-Sys (2022)) | × | 0.76B | 12.76 | 30.27 | 0.303 | 229 | – |
| BK–SDM–Base (Kim et al. (2024)) | × | 0.76B | 14.71 | 31.93 | 0.314 | 0.22 | 13 |
| LD–Pruner$^\star$ (Zhang et al. (2024b)) | × | 0.71B | 12.37 | 35.77 | 0.289 | 0.22 | – |
| SlimDiff (Ours, v1.5) | ✓ | 0.76B | 13.12 | 32.61 | 0.319 | 0.0005 | 4 |
| SlimDiff (Ours, v1.4) | ✓ | 0.76B | 13.21 | 31.96 | 0.289 | 0.0005 | 4 |
| *Autoregressive baselines* | | | | | | | |
| DALL-E (Ramesh et al. (2021)) | × | 12B | 27.5 | 17.9 | – | 250 | 8334 |
| CogView (Ding et al. (2021)) | × | 4B | 27.1 | 18.2 | – | 30 | – |

Table 2: **Efficiency comparison on MS-COCO** ($512 \times 512$, 50 steps, CFG= 8). MACs are reported per image generation. UNet (1): one U-Net forward pass; Whole (50): $50 \times$UNet (1). Latency measured with batch size $= 1$, fp16 for GPU, fp32 for CPU, and an identical scheduler.

| Model | Params (B) | MACs | | GPU Latency (s) | | CPU Latency (s) | |
|---|---|---|---|---|---|---|---|
| | | UNet(1) | Whole(50) | UNet(1) | Whole(50) | UNet(1) | Whole(50) |
| SD v1.5 | 1.04 | 169.5G | 8.5T | 0.032 | 1.57 | 1.90 | 85.60 |
| BK-SDM-Base | 0.76 | 112.0G | 5.6T | 0.020 | 0.92 | 0.77 | 34.56 |
| Small Stable Diffusion | 0.76 | 112.0G | 5.6T | 0.019 | 0.84 | 0.85 | 38.21 |
| **SlimDiff (Ours)** | **0.76** | **112.0G** | **5.6T** | **0.019** | **0.87** | **0.92** | **42.30** |

tage over competing methods that require retraining, such as BK-SDM-Base (13 A100 days) and Small Stable Diffusion (extensive retraining on 229M samples). Note that, for LD-Pruner, results are taken from the paper since the code is not open-sourced, and may not be directly comparable. Figure 3 demonstrates SlimDiff's ability to preserve semantic content and visual quality across diverse prompts. For the "cute Shiba Inu in a cabbage" prompt, SlimDiff maintains the key semantic elements: the dog's distinctive features and the cabbage setting, while achieving 27% parameter reduction (760M vs. 1.04B). Artistic prompts like "Van Gogh Starry Night" maintain characteristic brushstrokes and color palettes, while samples for prompts "man staring ahead" and "astronaut bears" highlight robustness across photorealistic and creative domains. More results in App. A.8

**3.2 How efficient is SlimDiff compared to baselines?** SlimDiff not only preserves generational quality but also delivers substantial efficiency gains. We assess efficiency along two axes: *inference cost* (MACs, GPU/CPU latency) and *training cost*. As shown in Table 2, SlimDiff reduces U-Net MACs by 34% per forward pass (112.0G vs. 169.5G) and likewise for full 50-step generation (5.6T vs. 8.5T). This translates into a 45% end-to-end GPU latency reduction (0.87s vs. 1.57s; $\sim 1.8\times$ faster) and a 51% CPU reduction (42.3s vs. 85.6s; $\sim 2.0\times$ faster). On training cost (Table 1), unlike baselines that require massive retraining, SlimDiff performs compression *without retraining* using only 500 **prompts and 4 A100-days** - over three orders of magnitude lighter than training from scratch, and dramatically smaller than data-hungry baselines.

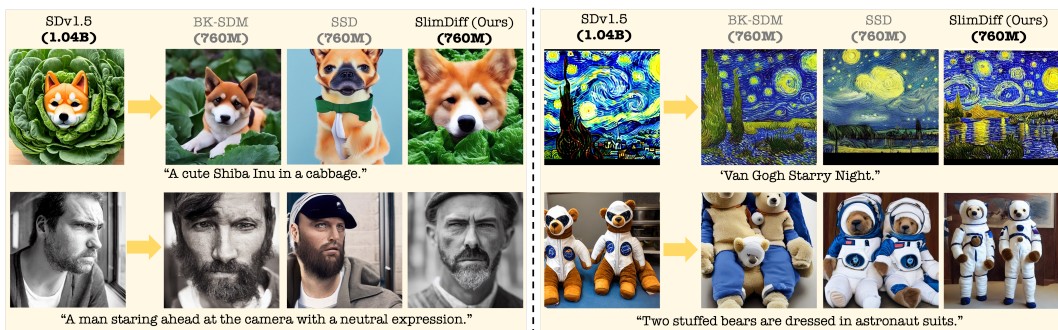

Figure 3: Visual comparison with contemporaries shows that SlimDiff maintains higher perceptual quality post-compression. Methods that rely on BP for model slimming are grayed out.

Table 3: Evaluation on human preference metrics (higher is better).

| Dataset | Model | Params | Score |
|---|---|---|---|
| HPS v2.1 | SD v1.5 | 1.04B | 24.45 |
| | SlimDiff (Ours) | 0.76B | 24.41 |
| ImageReward | SD v1.5 | 1.04B | 0.51 |
| | SlimDiff (Ours) | 0.76B | 0.56 |
| Pick-a-Pic v1 | SD v1.5 | 1.04B | 21.30 |
| | SlimDiff (Ours) | 0.76B | 21.22 |

Table 4: Cross-dataset CLIPScore. Rows denote calibration datasets; columns denote evaluation datasets, with diagonals showing in-domain performance.

| Calib → Eval | COCO | LAION | IRDB | Parti |
|---|---|---|---|---|
| COCO | **0.302** | 0.309 | 0.305 | 0.301 |
| LAION | 0.319 | **0.323** | 0.308 | 0.299 |
| IRDB | 0.301 | 0.311 | **0.314** | 0.300 |
| Parti | 0.300 | 0.309 | 0.307 | **0.303** |

**3.3 How transferable is SlimDiff across datasets?** A critical question for practical deployment is whether SlimDiff's compression strategy generalizes across different datasets and domains. To assess this transferability, we conduct cross-dataset evaluation experiments where we calibrate SlimDiff on one dataset and evaluate its performance on different target datasets, in Table 4. We consider four diverse benchmarks: MS-COCO ( Lin et al. (2014)) (natural scene descriptions), LAION ( Schuhmann et al. (2022)) (web-scale image–text pairs), IRDB ( Xu et al. (2023)) (human preference data), and PartiPrompts ( Yu et al. (2023)) (challenging compositional prompts). This setup probes whether SlimDiff's semantic slimming captures patterns that remain robust when transferred to new distributions. The strong cross-dataset results highlight SlimDiff's practical value: a model calibrated once on a readily available dataset such as COCO can be deployed across diverse domains without repeating the compression process. This makes the approach especially useful when target-domain data is limited, while also indicating that SlimDiff captures broad semantic structures that transfer reliably across different visual and textual distributions.

**3.4 Are Human Preference Metrics Robust to SlimDiff?** Beyond standard image quality metrics like FID and CLIP score, we evaluate SlimDiff's performance on human preference metrics to ensure that compression does not compromise perceptual quality or aesthetic appeal. We assess three established human preference benchmarks, using their own scoring methods: HPS v2.1 (holistic preference scoring) ( Wu et al. (2023)), ImageRewardDB ( Xu et al. (2023)) (reward-based preference), and Pick-a-Pic v1 (pairwise preference comparisons) ( Kirstain et al. (2023)).

Table 3 confirms that SlimDiff preserves alignment with human preferences despite a 27% parameter reduction. Across three distinct benchmarks, SlimDiff delivers similar scores to SD v1.5 on HPS v2.1 and Pic-a-Pic, while achieving a clear improvement on ImageReward. This consistency shows that our compression strategy not only maintains subjective quality but can also strengthen alignment with human judgments, underscoring SlimDiff's reliability for practical, user-facing deployment.

## 4 ABLATION

In this section, we ask: *How do design choices affect performance?* We ablate SlimDiff's core components to identify which factors drive quality and efficiency. Specifically, we study (i) SlimSet size, (ii) weighting strategies for timestep correlations, and (iii) the impact of compressing different module types. These analyses clarify why SlimDiff works and where its efficiency gains arise. Beyond these core ablations, Appx. 7 analyzes SlimSet coverage across datasets, Appx. 10 details per-block compression strategies, and Appx. 12 reports SlimDiff's extension to quantized baselines; we collate the main takeaways from these additional studies under (iv) *Extended Empirical Studies*.

**(i) SlimSet Size and Calibration Efficiency.** The size of the calibration set determines how well activation statistics are captured. Sets of very few prompts underrepresent semantic diversity and lead to performance drop, while larger sets yield diminishing returns beyond a certain size. Ablating SlimSet sizes from 500 up to 5,000 prompts in Fig. 3 show that around 500 prompts are already sufficient to match the quality of using tens of thousands, offering an effective balance between cost and fidelity. Different 500-prompt SlimSets produce near-identical activation statistics: the subspace overlap between these SlimSets is $\approx 1.0$. We refer to this property 'Stability', which ensures the calibration statistics are insensitive to a particular prompt sample - yielding reproducible, deployment-robust models rather than artifacts of one random subset. This effect is consistent across LAION-2B, COCO, PartiPrompts, and ImageRewardDB, confirming that a 500 prompt SlimSet provides a robust and stable calibration set. Accordingly, we adopt 500 as the standard SlimSet size.

Table 5: Ablation on weighting strategies for timestep-aware correlation accumulation. Lower FID is better.

| Weighting Strategy | FID↓ |
|---|---|
| Uniform over steps | 17.90 |
| Input activation diversity | 14.55 |
| Combined (diversity + weights) | 13.28 |
| **Spectral Influence Score (Ours)** | **13.12** |

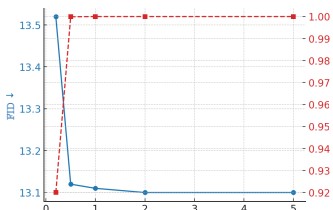

**Figure 3:** FID and stability as a function of SlimSet size.

Table 6: Module ablation on SD v1.5 using MS-COCO. ✓ = Compressed; × = Uncompressed. Quality degrades most with $\mathcal{FFN}$ compression

| SA | CA | $\mathcal{FFN}$ | FID↓ | CLIP↑ |
|---|---|---|---|---|
| ✓ | ✓ | × | 13.14 | 0.317 |
| ✓ | × | ✓ | 13.18 | 0.318 |
| × | ✓ | ✓ | 13.26 | 0.317 |
| ✓ | ✓ | ✓ | 13.12 | 0.319 |

**(ii) Weighting Strategies for Timestep-Aware Correlation Accumulation.** Not all timesteps contribute equally to perceptual fidelity. To evaluate our weighting design for accumulating correlations $\mathcal{C}_{l,t} \rightarrow \mathcal{C}_l$, we test four alternatives: (i) uniform averaging across timesteps, (ii) input-activation diversity only, (iii) spectral influence weighting (ours), and (iv) a combined scheme. As shown in Table 5, uniform weighting performs worst, while spectral influence yields the best FID (13.12). The combined scheme slightly improves over diversity-only but still falls short of spectral influence. These results highlight that fidelity-aware weighting, captured by the spectral influence score, is essential for effective timestep-aware accumulation of correlation matrices.

**(iii) Module-Specific Compression Contributions.** We ablate compression across self-attention (SA), cross-attention (CA), and feedforward ($\mathcal{FFN}$) modules to assess their relative contributions (Table 6).Note that, if a particular module is uncompressed, the compression budget is distributed over compressed modules. As our Nyström approximation for $\mathcal{FFN}$s constrains both gate and linear projections to share the same column selection matrix and relies on intermediate activations that are altered by compression, it creates architectural bottlenecks and circular dependencies that make $\mathcal{FFN}$ compression the most sensitive to quality loss. CA shows the next largest impact, while SA remains the most robust. Importantly, compressing all three modules jointly yields the best trade-off between quality and efficiency, highlighting the need for module-aware rather than uniform strategies.

**(iv) Extended Empirical Studies.** We (a) extend SlimDiff to FP16/INT8 post-training quantized SD v1.5, showing that SlimDiff+INT8 preserves MS-COCO quality while achieving up to $4N\times$ effective compression (Appx. 12); (b) compare against a spectrum of blockwise-slimming baselines (naive / Joint SVD, magnitude, PCA, Nova ( Nova et al. (2023)), SVD-LLM( Wang et al. (2024b))) and find that SlimDiff attains the lowest reconstruction error at matched ratios, and (c) validate SlimSet coverage across four datasets, where SlimSet covers 70–95% of quantile bins, and captures both common and rare prompts (Appx. 7).

**Future Work.** SlimDiff is extensible to DiT-style backbones such as Stable Diffusion 3.5's MMDiT (Appx. 11) via an RMSNorm-aware, data-driven QK decomposition tailored to jointly normalized query–key streams. A complementary direction is to refine propagation-aware rank allocation in joint attention, where image and text share heads but not FFNs, to more carefully balance cross-modal capacity under a single parameter budget.

## 5 CONCLUSION

We present **SlimDiff**, the first training-free framework that compresses diffusion models by aligning slimming with activation geometry. SlimDiff reduces the model by $\sim 100$M parameters and achieves up to 35% faster inference versus the original Stable Diffusion Models, all while consistently maintaining generation quality and human preference alignment across diverse benchmarks and datasets. Remarkably, it achieves these gains with only 500 calibration prompts – over $70\times$ fewer than prior work – and without any finetuning, through module-aware decompositions, timestep-weighted correlations, and a compact semantic coreset. Our analyses highlight three principles: effective compression depends more on functional structure than raw capacity, fidelity-aware weighting is critical to prevent error accumulation across timesteps, and module-aware strategies (especially cross-attention and feedforward) drive the best efficiency-quality trade-off. SlimDiff thus provides a principled, training-free compression alternative to retraining-based methods, demonstrating that Diffusion Models can be made both efficient and reliable without a single gradient step.

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

# A APPENDIX

## A.1 PROOFS

**Preliminaries:** $\mathcal{X}$ = input activation to the functional module; $\mathcal{W}_q, \mathcal{W}_k$ = query, key weight matrices; $\mathcal{W}_v, \mathcal{W}_o$ = value, output weight matrices; $\mathcal{W}_U$ = feedforward up-matrix , $\mathcal{W}_x, \mathcal{W}_g$ = part of the up-matrix divided into two, content matrix, gate matrix, $\mathcal{W}_D$ = feedforward down-matrix, $\mathrm{M}_k$ = k-column selection matrix

**Type-I: $\mathcal{FFN}$ via Nyström approximation Proof Sketch**

We perform FFN data-aware compression as described in Section 2.4. The module uses gated up-projections within a GeGLU nonlinearity followed by a down-projection. Since traditional low-rank compression of the joint matrices is ineffective, we constrain both up-projections to share a column selection matrix $\mathrm{M}_k$, reducing the objective to a Nyström approximation of the intermediate activation correlation matrix.

**What is a Column Selection Matrix?** A $k$-column selection matrix $\mathrm{M}_k \in \{0,1\}^{d \times k}$ has exactly one nonzero per column, indicating the selected indices. For any $A \in \mathbb{R}^{m \times d}$, the product $A\mathrm{M}_k$ consists of the $k$ selected columns of $A$. ( Lin et al. (2024); Wang & Singh (2018)). Note that, for any input matrix $X$ and any column selection matrix $\mathrm{M}$, the nonlinearity $\sigma$ satisfies $\sigma(X)\,\mathrm{M} = \sigma(X\mathrm{M})$. In other words, column selection commutes with $\sigma$. This assumption holds for all activation functions that act elementwise ( Pourkamali-Anaraki & Becker (2019); Gittens & Mahoney (2016)).

**Theorem 1 Proof Details** We constrain $\mathcal{W}_x, \mathcal{W}_g \in \mathcal{W}_U$ to be of the form $\mathcal{W}_U\mathrm{M}_k$. We define FFN intermediate activation $\mathcal{Z} = f(\mathcal{X}_i[\mathcal{W}_x|\mathcal{W}_g], \mathcal{W}_D) = (\mathcal{X}_i\mathcal{W}_x) \odot \sigma(\mathcal{X}_i\mathcal{W}_g)$, empirical correlation matrix of intermediate FFN features $\mathcal{K} = \mathcal{Z}^\top \mathcal{Z}$.

We can simplify equation 5 as:

$$\min_{\mathrm{M}_k, \widehat{\mathcal{W}}_D} \sum_{i=1}^{N} \left\| f(\mathcal{X}_i; [\mathcal{W}_x|\mathcal{W}_g], \mathcal{W}_D) - f(\mathcal{X}_i; [\mathcal{W}_x\mathrm{M}_k|\mathcal{W}_g\mathrm{M}_k], \widehat{\mathcal{W}}_D) \right\|_F^2$$

$$= \min_{\mathrm{M}_k, \widehat{\mathcal{W}}_D} \sum_{i=1}^{N} \left\| \left((\mathcal{X}_i\mathcal{W}_x) \odot \sigma(\mathcal{X}_i\mathcal{W}_g)\right)\mathcal{W}_D - \mathcal{Z}_i\mathrm{M}_k\widehat{\mathcal{W}}_D \right\|_F^2$$

$$= \min_{\mathrm{M}_k, \widehat{\mathcal{W}}_D} \sum_{i=1}^{N} \mathrm{Tr}\left((\mathcal{W}_D - \mathrm{M}_k\widehat{\mathcal{W}}_D)^\top \mathcal{Z}_i^\top \mathcal{Z}_i (\mathcal{W}_D - \mathrm{M}_k\widehat{\mathcal{W}}_D)\right) \tag{10}$$

$$= \min_{\mathrm{M}_k, \widehat{\mathcal{W}}_D} \left\| \left(\sum_{i=1}^{N} \mathcal{Z}_i^\top \mathcal{Z}_i\right)^{1/2}(\mathcal{W}_D - \mathrm{M}_k\widehat{\mathcal{W}}_D) \right\|_F^2$$

$$= \min_{\mathrm{M}_k, \widehat{\mathcal{W}}_D} \left\| \left(\mathcal{K}^{1/2}(\mathcal{W}_D - \mathrm{M}_k\widehat{\mathcal{W}}_D)\right） \right\|_F^2,$$

Note, we use the following properties from column section matrix for our equation- *(i) Column selection commutes with elementwise nonlinearities:$\sigma(\mathcal{X}_i\mathcal{W}_g\,\mathrm{M}_k) = \sigma(\mathcal{X}_i\mathcal{W}_g)\,\mathrm{M}_k$. (ii) Column extraction for linear terms: $\mathcal{X}_i\mathcal{W}_x\,\mathrm{M}_k = (\mathcal{X}_i\mathcal{W}_x)\,\mathrm{M}_k$. (iii) Columnwise compatibility of the Hadamard product with a shared selector: $\left((A\,\mathrm{M}_k) \odot (B\,\mathrm{M}_k)\right) = (A \odot B)\,\mathrm{M}_k$.*

**Optimal Down-Projection.** Setting the gradient of Eq. 10 with respect to $\widehat{\mathcal{W}}_D$ to zero yields the normal equations: $\mathrm{M}_k^\top \mathcal{K}\,\mathrm{M}_k\,\widehat{\mathcal{W}}_D = \mathrm{M}_k^\top \mathcal{K}\,\mathcal{W}_D$. The minimum-norm solution is therefore:

$$\widehat{\mathcal{W}}_D^\star = (\mathrm{M}_k^\top \mathcal{K}\,\mathrm{M}_k)^\dagger\,\mathrm{M}_k^\top \mathcal{K}\,\mathcal{W}_D. \tag{11}$$

**Reduction to Nyström Approximation.** Plugging Eq. 11 back into Eq. 10, we obtain

$$\min_{\mathrm{M}_k} \left\| \left(\mathcal{K}^{1/2} - \mathcal{K}^{1/2}\mathrm{M}_k(\mathrm{M}_k^\top \mathcal{K}\,\mathrm{M}_k)^\dagger\mathrm{M}_k^\top \mathcal{K}\right)\mathcal{W}_D \right\|_F^2 = \min_{\mathrm{M}_k} \|\mathcal{W}_D\|_2^2 \|\mathcal{K}^{-1/2}\|_2^2 \left\| \left(\mathcal{K} - \mathcal{K}\mathrm{M}_k(\mathrm{M}_k^\top \mathcal{K}\,\mathrm{M}_k)^\dagger\mathrm{M}_k^\top \mathcal{K}\right) \right\|_F^2 \tag{12}$$

$$\leq \|\mathcal{W}_D\|_2^2\,\|\mathcal{K}^{-1}\|_2\,E_{\mathrm{Nys}}^2(\mathcal{K}) \tag{13}$$

where $E_{\mathrm{Nys}}(\mathcal{K})$ denotes the Nyström approximation error ( Gittens & Mahoney (2016) of $\mathcal{K}$ using the same column selection $\mathrm{M}_k$.

While our derivation in Eq. 10 holds for any column selection strategy, in practice we adopt CPQR to construct $\mathrm{S}_k$. Strong rank-revealing QR Gu & Eisenstat (1996) ensures that the selected columns span a well-conditioned rank-$k$ subspace of $\mathcal{K}$, with Nyström reconstruction error bounded by a modest multiple of the optimal low-rank approximation error Gittens & Mahoney (2016).

**Type-II: $\mathcal{QK}$ via Whitening-SVD Proof Sketch**

We now turn to the query–key bilinear operator. We compress it via *whitening SVD*, which first rescales queries and keys by their input activation correlation roots, then applies SVD to the whitened cross-product. This yields the optimal rank-$r$ approximation of $\mathcal{W}_q \mathcal{W}_k^\top$ under the correlation-normalized Frobenius norm.

**Theorem 2 Proof Details:** For input $\mathcal{X}$, the $\mathcal{QK}$ interaction is $f(\mathcal{X}; \mathcal{W}_q, \mathcal{W}_k) = (\mathcal{X}\mathcal{W}_q)(\mathcal{W}_k^\top \mathcal{X}^\top)$, with rank at most $\min(d_q, d_k)$. We whiten queries and keys using their correlation roots $\mathcal{C}_q^{1/2}, \mathcal{C}_k^{1/2}$, apply SVD to the whitened cross-product, and truncate to rank $r$ (Eckart–Young–Mirsky). Unwhitening gives the compressed matrices: $\widehat{\mathcal{W}}_q = \mathcal{C}_q^{-1/2}\mathcal{U}_r, \qquad \widehat{\mathcal{W}}_k = \mathcal{C}_k^{-1/2}\mathcal{V}_r \Sigma_r$.

We can obtain Eq 5 as:

$$\min_{\widehat{\mathcal{W}}_q, \widehat{\mathcal{W}}_k} \sum_{i=1}^{N} \big\| f(\mathcal{X}_i; \mathcal{W}_q, \mathcal{W}_k) \ - \ f(\mathcal{X}_i; \widehat{\mathcal{W}}_q, \widehat{\mathcal{W}}_k) \big\|_F^2$$

$$= \min_{\widehat{\mathcal{W}}_q, \widehat{\mathcal{W}}_k} \sum_{i=1}^{N} \big\| (\mathcal{X}_i \mathcal{W}_q)(\mathcal{W}_k^\top \mathcal{X}_i^\top) \ - \ (\mathcal{X}_i \widehat{\mathcal{W}}_q)(\widehat{\mathcal{W}}_k^\top \mathcal{X}_i^\top) \big\|_F^2$$

$$= \min_{\widehat{\mathcal{W}}_q, \widehat{\mathcal{W}}_k} \sum_{i=1}^{N} \big\| \mathcal{X}_i (\mathcal{W}_q \mathcal{W}_k^\top - \widehat{\mathcal{W}}_q \widehat{\mathcal{W}}_k^\top) \mathcal{X}_i^\top \big\|_F^2 \tag{14}$$

$$= \min_{\widehat{\mathcal{W}}_q, \widehat{\mathcal{W}}_k} \big\| \underbrace{\Big(\sum_{i=1}^{N} \mathcal{X}_i^\top \mathcal{X}_i\Big)^{1/2}}_{\mathcal{C}_q^{1/2}} (\mathcal{W}_q \mathcal{W}_k^\top - \widehat{\mathcal{W}}_q \widehat{\mathcal{W}}_k^\top) \underbrace{\Big(\sum_{i=1}^{N} \mathcal{X}_i^\top \mathcal{X}_i\Big)^{1/2}}_{\mathcal{C}_k^{1/2}} \big\|_F^2$$

$$= \min_{\mathrm{rank} \leq r} \big\| \mathcal{C}_q^{1/2} \mathcal{W}_q \mathcal{W}_k^\top \mathcal{C}_k^{1/2} \ - \ \mathcal{C}_q^{1/2} \widehat{\mathcal{W}}_q \widehat{\mathcal{W}}_k^\top \mathcal{C}_k^{1/2} \big\|_F^2.$$

**Optimal Query, Key matrices.** Let $\Delta = \mathcal{W}_q \mathcal{W}_k^\top - \widehat{\mathcal{W}}_q \widehat{\mathcal{W}}_k^\top$. From Eq. 14 we have

$$\mathcal{V}_{\mathrm{II}} \ = \ \big\| \mathcal{C}_q^{1/2} \Delta \mathcal{C}_k^{1/2} \big\|_F^2 \ = \ \mathrm{Tr}\big((\mathcal{C}_q^{1/2}\Delta\mathcal{C}_k^{1/2})^\top(\mathcal{C}_q^{1/2}\Delta\mathcal{C}_k^{1/2})\big).$$

Differentiating w.r.t. $\widehat{\mathcal{W}}_q$ and $\widehat{\mathcal{W}}_k$ and setting to zero yields the normal equations:

$$\mathcal{C}_q \widehat{\mathcal{W}}_q (\widehat{\mathcal{W}}_k^\top \mathcal{C}_k \widehat{\mathcal{W}}_k) = \mathcal{C}_q \mathcal{W}_q \mathcal{W}_k^\top \mathcal{C}_k \widehat{\mathcal{W}}_k, \quad \mathcal{C}_k \widehat{\mathcal{W}}_k (\widehat{\mathcal{W}}_q^\top \mathcal{C}_q \widehat{\mathcal{W}}_q) = \mathcal{C}_k \mathcal{W}_k \mathcal{W}_q^\top \mathcal{C}_q \widehat{\mathcal{W}}_q. \tag{15}$$

Introducing whitened variables: $A = \mathcal{C}_q^{1/2}\widehat{\mathcal{W}}_q$, $B = \mathcal{C}_k^{1/2}\widehat{\mathcal{W}}_k$, and $\mathcal{M} = \mathcal{C}_q^{1/2}\mathcal{W}_q \mathcal{W}_k^\top \mathcal{C}_k^{1/2}$, the system becomes: $A(B^\top B) = \mathcal{M}B, \quad B(A^\top A) = \mathcal{M}^\top A$.

For the SVD $\mathcal{M} = U\Sigma V^\top$, the minimum-norm solution is $A^\star = U_r$, $B^\star = V_r \Sigma_r$, yielding the optimal rank-$r$ approximation $U_r \Sigma_r V_r^\top$. Unwhitening gives:

$$\widehat{\mathcal{W}}_q^\star = \mathcal{C}_q^{-1/2}U_r, \quad \widehat{\mathcal{W}}_k^\star = \mathcal{C}_k^{-1/2}V_r \Sigma_r. \tag{16}$$

The Type-II reconstruction error is bounded by the spectral tail:

$$\mathcal{V}_{II} \leq \sum_{i=r+1}^{\min(d_q, d_k)} \sigma_i^2(\mathcal{C}_q^{1/2}\mathcal{W}_q \mathcal{W}_k^\top \mathcal{C}_k^{1/2}). \tag{17}$$

**Type-III: VO via Whitening-SVD Proof Sketch**

We next analyze the value–output module, which, unlike QK, requires whitening only on the value side. The objective reduces to approximating the composite $\mathcal{W}_v\mathcal{W}_o$ under the metric induced by the input correlation matrix $\mathcal{C} = \sum_{i=1}^{N} \mathcal{X}_i^{\top}\mathcal{X}_i$.

**Theorem 3 (Adapted from ModeGPT( Lin et al. (2024))):** ModeGPT introduced a correlation-aware formulation showing that, under the metric defined by $\mathcal{C}$, the optimal low-rank approximation of the composite $\mathcal{W}_v\mathcal{W}_o$ is obtained by truncating the SVD of the whitened operator $\mathcal{C}^{1/2}\mathcal{W}_v\mathcal{W}_o$.

**Diffusion-Specific Adaptation:** Extending this principle to diffusion models, we specialize the modular objective in Eq. 5 for the $\mathcal{VO}$ module. Here, the coupling between value and output projections differs across attention types: in *self-attention*, both $\mathcal{W}_v$ and $\mathcal{W}_o$ operate on latent features, whereas in *cross-attention*, $\mathcal{W}_v$ originates from text-conditioning embeddings that interact with latent activations through $\mathcal{W}_o$. Under this setup, our optimization objective becomes:

$$\min_{\text{rank} \leq r} \left\| \mathcal{C}^{1/2}\mathcal{W}_v\mathcal{W}_o - \mathcal{C}^{1/2}\widehat{\mathcal{W}_v}\widehat{\mathcal{W}_o} \right\|_F^2, \tag{18}$$

whose optimal solution follows directly from the truncated SVD of $\mathcal{C}^{1/2}\mathcal{W}_v\mathcal{W}_o$, yielding

$$\widehat{\mathcal{W}_v} = \mathcal{C}^{-1/2}U_r, \qquad \widehat{\mathcal{W}_o} = \Sigma_r V_r^{\top},$$

and reconstruction error

$$\sum_{i=r+1} \sigma_i^2(\mathcal{C}^{1/2}\mathcal{W}_v\mathcal{W}_o).$$

## A.2 AUTOMATIC RANK ALLOCATION ENGINE (EXTENDED)

Under a fixed parameter budget $B$, we must distribute ranks $\{r_\ell\}_{\ell=1}^L$ across blocks to maximize compression fidelity. This constitutes a constrained optimization problem: maximize utility subject to $\sum_\ell c_\ell(r_\ell) \leq B$, where $c_\ell(\cdot)$ represents the cost model for block $\ell$. Directly estimating utility curves for each block is computationally prohibitive, requiring extensive sensitivity analysis across rank choices.

Instead, we leverage our trace-normalized Rayleigh quotient (TRQ) scores $\{s_\ell\}$ as tractable surrogates for block importance. TRQ captures how well each block's weights align with the dominant directions of their input activations. The trace normalization provides scale invariance across layers, while optional family-specific offsets can be added to $s_\ell$ (boosting $\mathcal{FFN}$ or cross-attention) to reflect their empirically higher contribution to fidelity within diffusion architectures.

**Convex Surrogate Formulation.** Let $\rho_\ell \in [0,1]$ denote the retention fraction (preserved rank relative to effective width) and $\phi_\ell = 1 - \rho_\ell$ the sparsity level. Following the entropy-regularized allocation framework of Lin et al. (2024), we solve:

$$\min_{\{\phi_\ell \in [0,1]\}} \sum_{\ell=1}^L \left( s_\ell\,\phi_\ell + \varepsilon\,\phi_\ell \log \phi_\ell \right) \quad \text{s.t.} \quad \frac{1}{L}\sum_{\ell=1}^L \phi_\ell = \bar{\phi} \tag{19}$$

where $\bar{\phi} \in [0,1]$ is the target average sparsity (determined by budget $B$) and $\varepsilon > 0$ is a temperature parameter. The linear term $s_\ell\phi_\ell$ penalizes sparsifying high-importance blocks as well as the early denoising steps (error accumulation-aware), while the entropy regularizer $\phi_\ell \log \phi_\ell$ prevents winner-take-all collapse by encouraging smooth allocation.

**Closed-Form Solution.** Problem equation 19 is strictly convex since $\phi \mapsto \phi \log \phi$ has positive second derivative. The Lagrangian optimality conditions yield:

$$s_\ell + \varepsilon(1 + \log \phi_\ell) + \frac{\lambda}{L} = 0$$

Solving and applying the sparsity constraint gives the unique softmax solution:

$$\phi_\ell = L\,\bar{\phi} \cdot \frac{\exp(-s_\ell/\varepsilon)}{\sum_{j=1}^L \exp(-s_j/\varepsilon)}, \qquad \rho_\ell = 1 - \phi_\ell \tag{20}$$

This exponential weighting automatically concentrates capacity on blocks with high TRQ scores and early denoising step-aware while maintaining smooth allocation controlled by temperature $\varepsilon$.

**Rank Mapping and Budget Enforcement.** Each block has effective width $d_\ell^{\text{eff}}$ ($d$ for $\mathcal{QK}/\mathcal{VO}$ per head, $4d$ for $\mathcal{FFN}$ intermediates). We convert retention fractions to hardware-friendly ranks:

$$r_\ell = \max\left\{r_{\min}, 8 \cdot \left\lfloor \frac{\rho_\ell d_\ell^{\text{eff}} + 4}{8} \right\rfloor \right\}$$

The rounding ensures tensor core alignment while $r_{\min} = 8$ prevents numerical instability.

Using cost model $c_\ell(r) = a_\ell r + b_\ell$ (where $a_\ell$ captures GEMM complexity), we find the target sparsity $\bar{\phi}$ via bisection search on the monotonic function $\bar{\phi} \mapsto \sum_\ell c_\ell(r_\ell(\bar{\phi}))$ until the total cost exactly meets budget $B$.

**Properties and Guarantees.** Our allocation is *convex* (unique global optimum), *propagation-aware* (high-influence blocks (in U-Net structure as well as in early denoising step) retain more capacity), and *budget-exact* (bisection ensures precise cost targeting). Unlike heuristic approaches, the entropy regularization provides principled smoothness while TRQ scores enable cross-family comparison without manual rescaling.

## A.3 RELATED WORK

We enumerate related works in this subsection, describing the popular structural pruning/factorization methods, alternative methods like reduction of timestep sampling or dynamic token pruning. We also show works on LLMs which use data-aware compression in their pipeline.

**Structural compression of diffusion models.** Several methods reduce the parameter footprint of diffusion models through pruning or architectural redesign. Block- and layer-pruning approaches compress the U-Net backbone but typically rely on distillation or finetuning to recover fidelity ( Kim et al. (2024); Zhang et al. (2024b;a)). Complementary efforts prune at the timestep or module granularity ( Fang et al. (2023); Yao et al. (2024a)), highlighting the role of sequential error propagation. Our work differs by providing a closed-form, training-free pipeline that operates over *functional groups* (QK, VO, FFN) with module-aligned objectives.

**Training-free acceleration at inference.** A separate thread accelerates sampling without changing model size. Attention-driven step reduction ( Wang et al. (2024a)) modulates compute over timesteps, while token and cache pruning seek runtime savings ( Bolya & Hoffman (2023)). These methods improve wall-clock latency but add per-run heuristics and do not permanently reduce parameters. SlimDiff is orthogonal: it permanently shrinks dimensions/ranks and can be combined with such inference-time techniques.

**Quantization for diffusion models.** Post-training quantization (PTQ) has been adapted to diffusion pipelines to reduce precision while preserving sample quality ( Zeng et al. (2025)). Recent works study timestep-aware calibration, noise-schedule sensitivity, and stability at low bit-widths; surveys synthesize progress and open challenges. Quantization is complementary to SlimDiff: the former reduces *precision*, while SlimDiff reduces *structure*; together they offer stacked efficiency gains.

**Activation-/data-aware compression and modular views.** Beyond diffusion, activation-aligned and module-aware compression in large transformers ( Lin et al. (2024); Ashkboos et al. (2024)) shows that respecting activation geometry and functional coupling outperforms naive matrix-wise dimension reduction. SlimDiff adapts this perspective to diffusion's evolving activations: it models per-timestep correlations, weights them by spectral influence, and applies whitening–SVD (QK/VO) and Nyström FFN reductions under a global rank allocator.

## A.4 EXPERIMENTAL SETUP

**Models and Code.** We evaluate on Stable Diffusion v1.5 and v1.4 using the publicly released U-Net, VAE, and text-encoder weights, and include two public compressed baselines: *BK-SDM-*

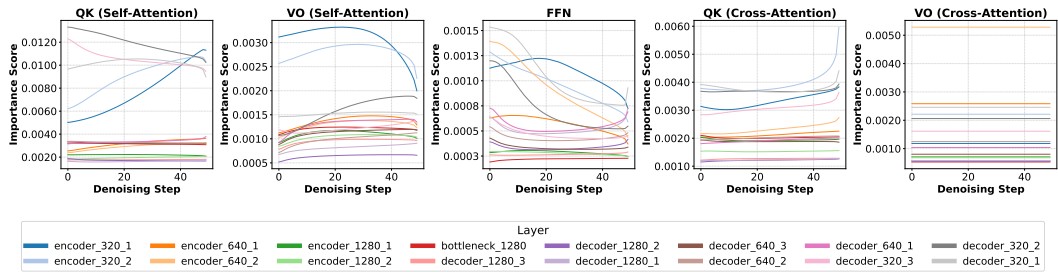

Figure 4: Spectral Influence Score Distribution across different functional modules

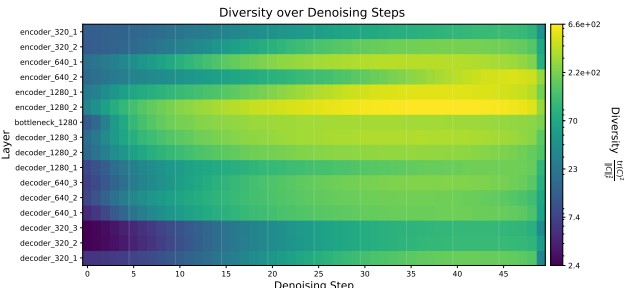

Figure 5: Diversity distribution of input activation across different functional modules

*Base* from nota-ai and *Small Stable Diffusion* from OFA-Sys (all checkpoints obtained from their HuggingFace model hubs).

**Datasets.** Unless noted, we report results on MS-COCO 2014 val at $512 \times 512$. For calibration, we construct a 500-prompt *SlimSet* by sampling text queries from LAION-Aesthetics V2 subset($\sim$212k pairs); only the text side is used. For cross-domain robustness and human preference evaluation, we additionally test with prompts from LAION-Aesthetics, COCO, PartiPrompts, and ImageRewardDB. The same SlimSet is reused across models/datasets with no per-dataset tuning.

**Implementation details.** All activation collection and compression/optimization procedures run on a single NVIDIA A100 80GB. Unless specified, inference uses **50** denoising steps of the UNet with classifier-free guidance **CFG**=8. We use the default latent resolution ($H = W = 64$) yielding $512 \times 512$ images, and keep scheduler and sampling settings fixed across experiments. Code is based on Diffusers and PyTorch, with minor utilities for data-aware factorization.

**Metrics.** Quality: FID (COCO), Inception Score, CLIPScore; Human preference: HPS v2.1, ImageReward, and Pick-a-Pic scoring on their own datasets. For cross-dataset evaluation, calibration and evaluation prompts come from different corpora as indicated in the main text.

**Latency and MACs.** MACs are reported per image for one UNet forward and for a full 50-step trajectory. GPU and CPU latencies use identical schedulers with batch size 1 and fp16(GPU)/fp32(CPU). All wall-clock measurements are averaged over multiple runs after a warm-up pass.

## A.5 SPECTRAL INFLUENCE SCORE AND DIVERSITY RESULTS ACROSS LAYERS AND MODULES

We measure the compression sensitivity of each block using the *trace-normalized Rayleigh quotient (TRQ)* influence score. TRQ evaluates how strongly a block's weight operator aligns with the dominant eigenspaces of its activation covariance. Intuitively, it quantifies how effectively a block exploits the signal-rich directions of its input. Because TRQ is normalized by the input trace, the

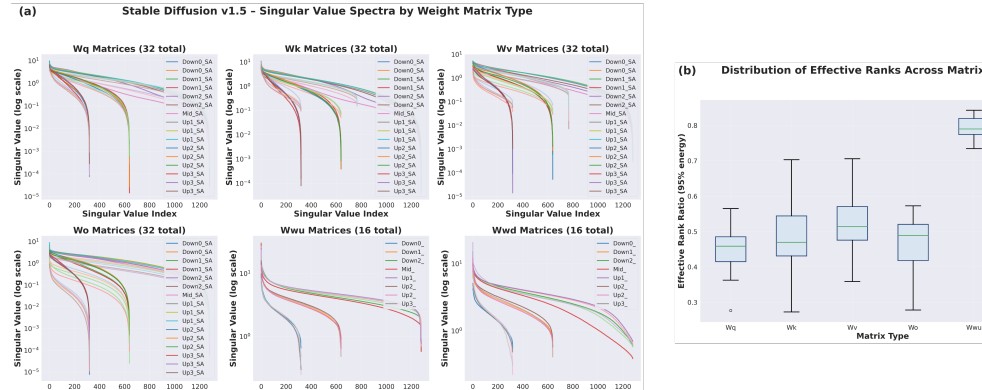

Figure 6: Singular value analysis of Stable Diffusion v1.5 weights. (a) Singular value spectra across attention and feedforward matrices reveal heavy-tailed distributions without clear low-rank cutoffs. (b) Effective rank ratios (95% energy) show that most weights remain in a high-rank regime for FFN layers, while QK/VO module weights average around 50% rank. Importantly, effective ranks must fall below 50% to yield meaningful compression gains.

score is directly comparable across blocks of different widths and even across module families ($\mathcal{QK}$, $\mathcal{VO}$, $\mathcal{FFN}$).

Figure 4 highlights several consistent trends. Across all module types, the widest layers (1280 channels) tend to have the lowest importance. For $\mathcal{QK}$ blocks, decoder layers dominate encoder layers, reflecting their exposure to greater variance. In contrast, $\mathcal{FFN}$ blocks show higher importance in the encoder, especially at early timesteps. For $\mathcal{VO}$, cross-attention blocks exhibit stable importance across timesteps, since they depend only on text-side correlations. These observations guide our compression strategy, where rank allocation is driven directly by TRQ.

We also examine *diversity*, defined as the spectral spread of input activations. Diversity is highest at early timesteps and decreases as denoising progresses, with mid-U-Net layers achieving higher diversity in fewer steps. This agrees with prior findings that greater diversity reflects less noisy, more semantically aligned activations Wang et al. (2024a). However, diversity alone does not reliably predict compression sensitivity. As confirmed in Table 5, TRQ provides a stronger and more actionable signal for structural rank reduction, while diversity remains useful primarily as a diagnostic measure.

A.6    SPECTRAL ANALYSIS OF SDM WEIGHTS

Most attention and feedforward weights in Stable Diffusion fall within a high-rank regime, as evident in Fig. 6. While 40-60% compressibility may appear possible, this is insufficient: reductions below ~50% have little impact on overall parameter count, while more aggressive rank reduction causes sharp quality degradation due to sequential error propagation across denoising steps ( Lin et al. (2024)).

Unlike prior works in diffusion models, that apply SVD independently to each weight matrix, we propose a *joint matrix decomposition* strategy. By respecting the functional couplings within attention (e.g., $W_q W_k^\top$, $W_v W_o$) and integrating data-aware statistics, our method achieves meaningful compression without sacrificing generation quality.

A.7    SLIMSET: ADDITIONAL METHODOLOGY DETAILS AND COVERAGE ANALYSIS

We provide additional details on SlimSet construction and empirical evidence that SlimSet is both statistically and semantically representative of the full prompt corpus.

**Construction methodology.**    As described in Sec. 2.2, SlimSet construction proceeds in two stages. **Stage 1** stratifies the corpus by distinctiveness: we embed all prompts with CLIP, compute

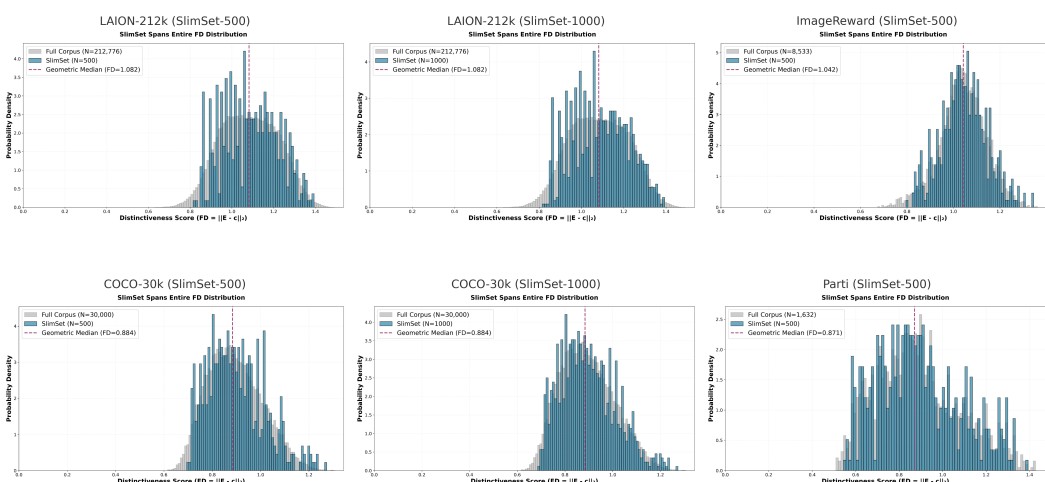

Figure 7: **SlimSet distinctiveness-score coverage.** Histograms of the full corpus (grey) and SlimSet (blue) for each dataset. SlimSets span the entire distribution and match the corpus median (dashed), confirming coverage of both common (low-$f$) and rare (high-$f$) patterns.

| Dataset | Corpus | SlimSet | $\% <$ median | $\% < p_{25}$ | Bins cov. | Medians |
|---|---|---|---|---|---|---|
| LAION-212K | 212,776 | 500 | 50.0% (250) | 25.0% (125) | 14/20 (70%) | 1.082 / 1.082 |
| LAION-212K | 212,776 | 1000 | 50.0% (500) | 25.0% (250) | 14/20 (70%) | 1.082 / 1.082 |
| COCO-30K | 30,000 | 500 | 50.0% (250) | 24.8% (124) | 17/20 (85%) | 0.884 / 0.884 |
| COCO-30K | 30,000 | 1000 | 50.0% (500) | 24.8% (248) | 17/20 (85%) | 0.884 / 0.884 |
| ImageReward-8K | 8,533 | 500 | 50.0% (250) | 25.2% (126) | 17/20 (85%) | 1.042 / 1.043 |
| PartiPrompts-1.6K | 1,632 | 500 | 50.0% (250) | 25.0% (125) | 19/20 (95%) | 0.871 / 0.871 |

Table 7: **SlimSet distinctiveness statistics across datasets.** For all corpora and both coreset sizes, exactly half of SlimSet samples lie below the corpus median and roughly one quarter lie in the lowest quartile, confirming that prompts near the embedding-space center are systematically included rather than discarded. SlimSets also match the corpus medians and cover 70-95% of the quantile bins, showing that they span both central and tail regions of the distribution.

the geometric median $c$ of the embedding cloud, and assign each prompt a score $f_i = \|E_i - c\|_2$. We then partition prompts into $B$ equal-mass *quantile bins* based on $\{f_i\}$. Low-$f$ bins contain prompts near the centroid (frequent, canonical patterns); high-$f$ bins contain rare or compositional prompts. Crucially, this stratification *preserves* low-score prompts rather than discarding them.

**Stage 2** performs diverse sampling within each bin. Given target size $J$, we assign each bin $b$ a quota $q_b$ (proportional to bin population) and apply farthest-point sampling (FPS) constrained to that bin. This yields a well-spread subset from *every* region of the $f$-distribution: center, middle, and tails alike. Finally, we union all per-bin subsets and apply cosine-based de-duplication to obtain the final SlimSet $\mathcal{S}$.

**Coverage analysis.** We analyse SlimSets constructed on four corpora (LAION-212K, COCO-30K, ImageReward-8K, PartiPrompts-1.6K) at sizes $J \in \{500, 1000\}$. Fig. 7 overlays the distinctiveness-score distributions of each full corpus (grey) and corresponding SlimSet (blue). In all cases, SlimSet spans the entire support and closely tracks the corpus distribution, including the low-$f$ region near the geometric median (dashed line).

**Quantitative statistics.** Table 7 reports coverage metrics for all configurations: the fraction of SlimSet prompts below the corpus median distinctiveness score and below the 25th percentile ($p_{25}$), the fraction of quantile bins containing at least one SlimSet sample, and the median $f$-values of the corpus and SlimSet. Across all six SlimSets, we observe a consistent $\approx 25/50/25$ split across the

| Dataset | Region | Example SlimSet prompts (distinctiveness score) |
|---------|--------|--------------------------------------------------|
| PartiPrompts | Low | "a clock" (0.56); "a mountain" (0.56); "a motorcycle" (0.57) |
| | High | "a portrait of a statue of a pharaoh wearing steampunk glasses, white t-shirt and leather jacket" (1.39) |
| COCO-30K | Low | "A guy in a suit and tie is posing for the camera" (0.69); "The boy is skating during the day" (0.70) |
| | High | "Citrus heights water district building with large orange sculptures and a flagpole" (1.27) |
| LAION-212K | Low | "California Hills And Vines Paintings" (0.82); "mountain village by nikolai dubovskoy" (0.83) |
| | High | "Flags On Faces Semmick Photo – Zephyr by Steve Henderson" (1.39) |
| ImageReward | Low | "a beautiful plant, aesthetic, oil painting, pale colors, high detail, 8k" (0.80) |
| | High | "a dezeen showroom, archdaily photo of synthesizers by virgil abloh & Patricia Urquiola" (1.34) |

Table 8: **Qualitative SlimSet examples by distinctiveness region.** Low-$f$ prompts are frequent, canonical patterns; high-$f$ prompts are rare or compositional. This confirms that SlimSet captures both cluster centres and diverse semantics across all datasets.

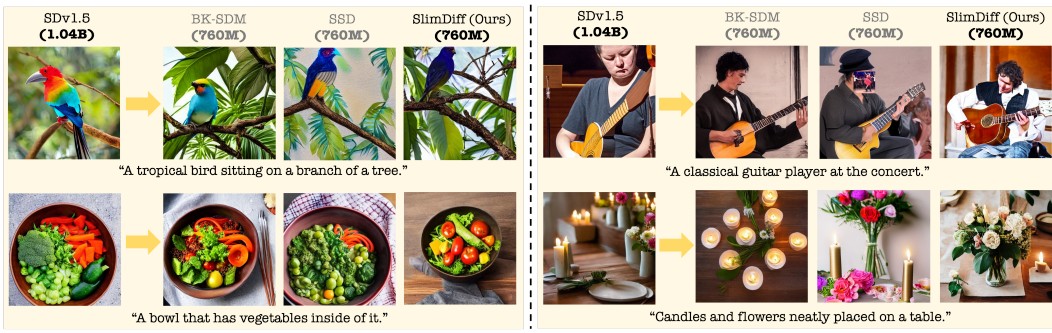

Figure 8: Additional visual comparison with contemporaries on SDv1.5 demonstrates that SlimDiff maintains higher perceptual quality post-compression. Methods that rely on BP for model slimming are grayed out.

lower quartile, interquartile range, and upper quartile of distinctiveness, with near-identical medians between SlimSet and full corpus. This behaviour is a direct consequence of quantile-based stratification: equal-mass bins combined with proportional quotas ensure that prompts near the geometric median (cluster centres) receive the same systematic coverage as tail-region prompts.

**Qualitative examples.** Table 8 lists representative SlimSet prompts from low- and high-$f$ regions. Low-$f$ entries are simple, canonical patterns (e.g., "a clock", "a mountain"), while high-$f$ entries are rare or compositional (steampunk scenes, multi-object descriptions). Together with the quantitative results above, these examples demonstrate that SlimSet captures both frequent cluster centres and diverse tail semantics.

## A.8 ADDITIONAL VISUAL RESULTS

As shown in Fig. 8 and Fig 9, SlimDiff consistently preserves generation quality while matching or surpassing structurally compressed baselines (BK-SDM, SSD) across diverse prompts. Importantly, this holds for both SDv1.5 and SDv1.4 backbones, demonstrating that our method is not tied to a particular model variant. SlimDiff thus achieves high fidelity under significant parameter reduction, highlighting its robustness and generality across architectures.

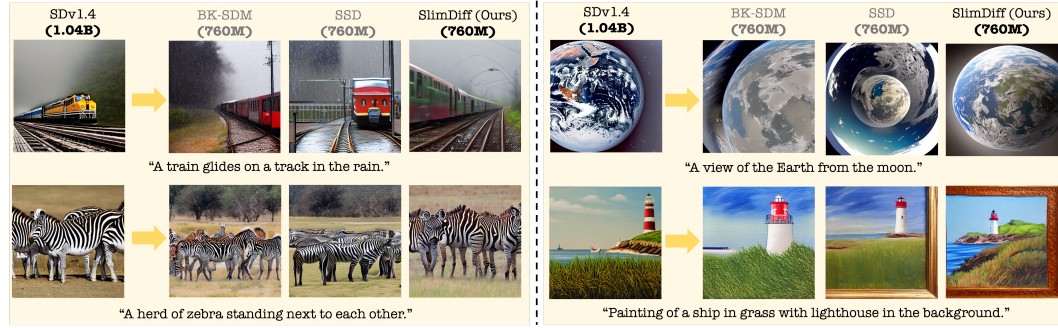

Figure 9: Qualitative comparison across baselines on SDv1.4 highlights SlimDiff's ability to retain generative performance under compression.

Table 9: Summary of key notation used throughout the paper.

| Symbol | Definition |
|---|---|
| *Models and Optimization* | |
| $\Theta, \hat{\Theta}$ | Original and compressed diffusion model parameters |
| $B$ | Target parameter budget |
| $\mathcal{L}_{\text{qual}}$ | Quality loss measuring output fidelity |
| *Activations and Correlations* | |
| $x_{l,t}^{(i)} \in \mathbb{R}^d$ | $i$-th activation sample (spatial location) at layer $l$, timestep $t$ |
| $\mathcal{X}_{l,t} \in \mathbb{R}^{N_t \times d}$ | Stacked activation matrix ($N_t$ spatial samples across prompts) |
| $N_t$ | Total number of spatial samples at layer $l$, timestep $t$ |
| $\widehat{\mathcal{C}}_{l,t}$ | Empirical second moment: $\frac{1}{N_t} \mathcal{X}_{l,t}^{\top} \mathcal{X}_{l,t}$ |
| $\widetilde{\mathcal{C}}_{l,t}$ | Regularized covariance: $\widehat{\mathcal{C}}_{l,t} + \epsilon \mathbf{I}$ ($\epsilon = 10^{-6}$) |
| $\bar{\mathcal{C}}_l$ | Timestep-aggregated correlation: $\sum_t w_{l,t} \widetilde{\mathcal{C}}_{l,t}$ |
| $\bar{\mathcal{R}}_l$ | Whitening transform: $\bar{\mathcal{C}}_l^{1/2}$ |
| *Scoring and Allocation* | |
| $\mathcal{I}_{l,t}$ | Spectral influence score (TRQ) at layer $l$, timestep $t$ |
| $w_{l,t}$ | Timestep weighting derived from $\mathcal{I}_{l,t}$ |
| $r_\ell, \rho_\ell, \phi_\ell$ | Rank, retention fraction, sparsity for block $\ell$ |
| *Weight Matrices* | |
| $\mathcal{W}_q, \mathcal{W}_k, \mathcal{W}_v, \mathcal{W}_o$ | Query, key, value, output projection matrices |
| $\mathcal{W}_x, \mathcal{W}_g, \mathcal{W}_D$ | FFN content, gate, and down-projection matrices |
| $\mathbf{M}_k$ | $k$-column selection matrix for Nyström approximation |

## A.9 NOTATION SUMMARY

We provide the notation summary for the paper in Table 9

## A.10 BLOCKWISE-SLIMMING BASELINE METHODS COMPARISON

This appendix details the baseline methods used in our blockwise-slimming ablation. We compare SlimDiff against a spectrum of rank-reduction approaches that vary along two design axes: (i) *data awareness* (whether input activations are used) and (ii) *module alignment* (whether the method compresses the joint computation, e.g., $W_q W_k^{\top}$, instead of individual matrices).

**Category 1: No Data Awareness (Weight-Only).**

**Naive SVD.** A per-matrix low-rank baseline. Each projection $W \in \{W_q, W_k, W_v, W_o\}$ is compressed independently via $W \approx U_k \Sigma_k V_k^{\top}$, where $k$ is the target rank. This ignores both module structure and input statistics.

**Joint SVD.** A module-aligned, data-agnostic baseline. For QK, we decompose the product

$$W_q W_k^{\top} \approx U_k \Sigma_k V_k^{\top},$$

and factor it as $W_q \approx U_k \Sigma_k^{1/2}$, $W_k \approx V_k^\top \Sigma_k^{1/2}$. Analogously for VO. This respects the joint computation of the attention block, but remains oblivious to activation statistics.

**Magnitude Pruning.** A simple structural pruning baseline that ranks channels by their weight norms:

$$\text{importance}_i = \|W_q[:,i]\|_2 + \|W_k[:,i]\|_2 + \|W_v[:,i]\|_2 + \|W_o[i,:]\|_2,$$

keeps the top-$k$ channels, and zeros out the rest. No data or module coupling is used.

**Category 2: Data-Aware, Not Module-Aligned.**

**PCA.** We use principal components of the input activations $X$ to define a data-aware projection subspace. First, we form the empirical correlation matrix $C = \frac{1}{N} X^\top X$ and compute its top-$k$ eigenvectors $V_k = eig_k(C)$. Each projection matrix is then compressed independently by projecting onto this subspace, e.g., $W_q \approx V_k V_k^\top W_q$ and $W_k \approx V_k V_k^\top W_k$. Thus PCA leverages activation geometry but still operates on $W_q$ and $W_k$ separately, without jointly modeling the QK product.

**Nova Nova et al. (2023).** Nova ranks channels using the activation variance, $\text{importance}_i = \text{Var}(X[:,i])$, and prunes by keeping the highest-variance channels, making it data-aware but still not jointly aligned across QK/VO.

**SVD-LLM Wang et al. (2024b).** This method applies whitening to activations, performs SVD in the whitened space, and then unwhitens the compressed weights. In our implementation it whitens each matrix separately, so it is data-aware but not strictly module-aligned.

**SlimDiff.** We perform a joint, data-aware low-rank decomposition of the effective attention computation: we whiten the QK (and VO) product using the input correlation $C$, apply SVD in the whitened space, and then unwhiten and refactor the result to obtain compressed $W_q, W_k$ (and $W_v, W_o$); for FFN, we use a CPQR+Nyström variant. This yields a module-aligned low-rank subspace that is explicitly adapted to the dominant modes of the activations.

**Experimental Setup.** Stable Diffusion v1.5 with 16 self-attention layers is used. We compress (i) attention QK and VO projections and (ii) FFN GEGLU up/down projections at four compression ratio levels: 95%, 90%, 75%, and 50% . Layer-wise reconstruction error is measured as relative Frobenius norm, Error $= \frac{\|Y - \hat{Y}\|_F^2}{\|Y\|_F^2}$, where $Y$ is the original layer output and $\hat{Y}$ is the compressed output. Activations are collected from 40 diverse prompts from PartiPrompts, spanning objects, scenes, and styles.

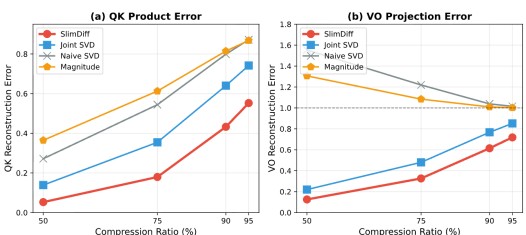

Figure 10: Output Error vs Compression Ratio

**Output error at 75% Compression Ratio.** Table 10 summarizes average reconstruction errors at 75% compression ratio. Across QK+VO+FFN, SlimDiff achieves the best average error (0.617), despite operating at the same compression ratio. We also show output error vs compression ration for an attention layer, in Fig 10. Since the FFN block includes a GEGLU nonlinearity rather than a purely linear operator, Joint SVD, Nova, and PCA are not directly applicable there, so their FFN entries are left blank in the table.

**Layerwise Behavior.**

Table 11 reports representative layers at 75% compression ratio.

SlimDiff's relative gains are largest in the mid_block bottleneck, where QK error at 75% retention drops from 0.209 (Joint SVD) to 0.099 (**53% reduction**). This suggests that aligning the low-rank subspace with activation geometry is particularly important in capacity-critical layers.

**Effect of Module Alignment.**

To isolate the role of module alignment at 75% compression:

Table 10: Layer-wise reconstruction error at 75% compression ratio. Lower is better. Methods are grouped by design properties. SlimDiff yields the lowest error for QK, VO, and FFN.

| Method | QK Error ↓ | VO Error ↓ | FFN Error ↓ | Avg. ↓ |
|---|---|---|---|---|
| *No data awareness* | | | | |
| Naive SVD | 0.544 | 1.220 | 2.679 | 1.481 |
| Joint SVD | 0.354 | 0.480 | – | 0.417 |
| Magnitude | 0.613 | 1.084 | 0.239 | 0.645 |
| *Data-aware, not module-aligned* | | | | |
| PCA | 0.873 | 1.011 | – | 0.942 |
| Nova | 0.613 | 1.078 | – | 0.846 |
| SVD-LLM | 0.447 | 1.304 | – | 0.876 |
| *Module-aligned + data-aware (ours)* | | | | |
| **SlimDiff** | **0.179** | **0.326** | **1.346** | **0.617** |

Table 11: Per-layer reconstruction error at 75% compression for representative layers. SlimDiff's gains are largest in bottleneck layers (mid_block).

| Layer Type | Method | QK Error | VO Error |
|---|---|---|---|
| Early (down_blocks.0) | SlimDiff | **0.190** | **0.336** |
| | Joint SVD | 0.345 | 0.446 |
| | PCA | 0.901 | 1.007 |
| | Magnitude | 0.556 | 1.077 |
| Middle (mid_block) | SlimDiff | **0.099** | **0.304** |
| | Joint SVD | 0.209 | 0.480 |
| | PCA | 0.808 | 1.014 |
| | Magnitude | 0.613 | 1.077 |
| Late (up_blocks.3) | SlimDiff | **0.043** | **0.230** |
| | Joint SVD | 0.233 | 0.461 |
| | PCA | 0.816 | 1.012 |
| | Magnitude | 0.465 | 1.090 |

| Configuration | QK Error (75%) | Reduction vs. PCA |
|---|---|---|
| PCA (data-aware, not aligned) | 0.873 | baseline |
| Joint SVD (aligned, no data) | 0.354 | 59% |
| SlimDiff (aligned + data) | **0.179** | **79%** |

Module alignment alone (PCA → Joint SVD) yields a larger improvement than data-awareness applied without alignment. Combining both, as in SlimDiff, gives the strongest reduction in error at the same compression ratio.

## A.11 EXTENSION TO DIFFUSION TRANSFORMERS: STABLE DIFFUSION 3.5

We extend SlimDiff to Stable Diffusion 3.5, which uses a Multi-Modal Diffusion Transformer (MMDiT) Stability AI (2024) with 24 blocks. Each block processes image tokens $\mathbf{x}_{\text{img}} \in \mathbb{R}^{N_{\text{img}} \times d_{\text{model}}}$ and text tokens $\mathbf{x}_{\text{txt}} \in \mathbb{R}^{N_{\text{txt}} \times d_{\text{model}}}$ through dual streams with joint attention and separate FFNs. Table 12 summarizes the resulting differences in compression strategy compared to SD v1.5.

**Joint Attention with RMS Normalization.** For each head $j$, the image and text streams project independently, apply RMS normalization to Q/K, and are then concatenated into a joint attention

| Component | SD v1.5 | SD 3.5 | Reason |
|---|---|---|---|
| Q/K method | Whitened SVD | CR | RMS norm |
| V/O method | Whitened SVD | Whitened SVD | No norm |

Table 12: Compression method comparison. Column selection (CR) accommodates RMS-normalized Q/K, while joint importance enforces dimensional coupling only where algebraically required.

space:

$$\mathbf{Q}_{\text{img}}^j = \mathbf{x}_{\text{img}}\mathbf{W}_{Q,\text{img}}^j, \quad \mathbf{K}_{\text{img}}^j = \mathbf{x}_{\text{img}}\mathbf{W}_{K,\text{img}}^j, \quad \mathbf{V}_{\text{img}}^j = \mathbf{x}_{\text{img}}\mathbf{W}_{V,\text{img}}^j,$$

$$\hat{\mathbf{Q}}_{\text{img}}^j = \text{RMSNorm}(\mathbf{Q}_{\text{img}}^j), \quad \hat{\mathbf{K}}_{\text{img}}^j = \text{RMSNorm}(\mathbf{K}_{\text{img}}^j),$$

and analogously for the text stream. Concatenating along the token dimension yields

$$\mathbf{Q}_{\text{joint}}^j = \begin{bmatrix} \hat{\mathbf{Q}}_{\text{img}}^j \\ \hat{\mathbf{Q}}_{\text{txt}}^j \end{bmatrix}, \quad \mathbf{K}_{\text{joint}}^j = \begin{bmatrix} \hat{\mathbf{K}}_{\text{img}}^j \\ \hat{\mathbf{K}}_{\text{txt}}^j \end{bmatrix}, \quad \mathbf{V}_{\text{joint}}^j = \begin{bmatrix} \mathbf{V}_{\text{img}}^j \\ \mathbf{V}_{\text{txt}}^j \end{bmatrix}. \tag{21}$$

With $d_{\text{head}}$ the per-head dimension, attention has block structure

$$\mathbf{A}^j = \text{softmax}\left( \frac{\mathbf{Q}_{\text{joint}}^j (\mathbf{K}_{\text{joint}}^j)^\top}{\sqrt{d_{\text{head}}}} \right) = \begin{bmatrix} \mathbf{A}_{\text{img}\to\text{img}}^j & \mathbf{A}_{\text{img}\to\text{txt}}^j \\ \mathbf{A}_{\text{txt}\to\text{img}}^j & \mathbf{A}_{\text{txt}\to\text{txt}}^j \end{bmatrix}. \tag{22}$$

Outputs are then split and projected back to their respective streams:

$$\text{attn\_out}_{\text{img}}^j = \mathbf{A}_{\text{img}\to\text{img}}^j \mathbf{V}_{\text{img}}^j + \mathbf{A}_{\text{img}\to\text{txt}}^j \mathbf{V}_{\text{txt}}^j, \quad \text{out}_{\text{img}}^j = \text{attn\_out}_{\text{img}}^j \mathbf{W}_{O,\text{img}}^j, \tag{23}$$

$$\text{attn\_out}_{\text{txt}}^j = \mathbf{A}_{\text{txt}\to\text{img}}^j \mathbf{V}_{\text{img}}^j + \mathbf{A}_{\text{txt}\to\text{txt}}^j \mathbf{V}_{\text{txt}}^j, \quad \text{out}_{\text{txt}}^j = \text{attn\_out}_{\text{txt}}^j \mathbf{W}_{O,\text{txt}}^j. \tag{24}$$

**Compression Strategy.** MMDiT introduces two structural challenges for SlimDiff: (i) *Dimensional coupling.* Because image and text tokens are concatenated and attend jointly, Q/K must share a common reduced dimension $k$ across both streams. (ii) *RMS normalization.* Correlations of Q/K are naturally defined on pre-normalized outputs, whereas attention operates on RMS-normalized $\hat{\mathbf{Q}}, \hat{\mathbf{K}}$. Any Q/K compression must therefore respect the normalization-induced geometry.

We address these via a hybrid strategy: column selection for Q/K (to respect RMSNorm and coupling), and standard whitened SVD for V/O (where streams can remain independent).

**Query–Key: Column Selection.** For Q/K with RMS normalization, we adopt a column selection scheme Drineas et al. (2006) based on correlation matrix norms. Let $\mathbf{C}_{Q,\text{img}}$ and $\mathbf{C}_{K,\text{img}}$ denote correlations of pre-normalized Q/K for the image stream in head $j$, and similarly for text. For dimension $i$ in head $j$ we define stream-wise scores

$$s_{\text{img},i}^j = \left\| \mathbf{C}_{Q,\text{img}}^{1/2}[:,i] \right\|_2 \cdot \left\| \mathbf{C}_{K,\text{img}}^{1/2}[:,i] \right\|_2, \quad s_{\text{txt},i}^j = \left\| \mathbf{C}_{Q,\text{txt}}^{1/2}[:,i] \right\|_2 \cdot \left\| \mathbf{C}_{K,\text{txt}}^{1/2}[:,i] \right\|_2. \tag{25}$$

A joint importance score aggregates both streams:

$$s_{\text{joint},i}^j = \frac{s_{\text{img},i}^j + s_{\text{txt},i}^j}{2}. \tag{26}$$

We select the top-$k$ dimensions $\mathcal{I}^j = \text{top-k}\{s_{\text{joint},i}^j\}$ and form a selection matrix $\mathbf{S}_k^j \in \mathbb{R}^{d_{\text{head}} \times k}$. Both streams are compressed using the same $\mathbf{S}_k^j$:

$$\mathbf{W}_{Q,\text{img}}^{j,c} = \mathbf{W}_{Q,\text{img}}^j \mathbf{S}_k^j, \quad \mathbf{W}_{K,\text{img}}^{j,c} = \mathbf{W}_{K,\text{img}}^j \mathbf{S}_k^j, \tag{27}$$

and analogously for the text stream. This preserves the algebraic requirement that $\hat{\mathbf{Q}}_{\text{joint}}^j (\hat{\mathbf{K}}_{\text{joint}}^j)^\top$ remains well-defined in the reduced space.

**Value–Output: Independent Whitened SVD.** Although $\mathbf{V}_{\text{img}}$ and $\mathbf{V}_{\text{txt}}$ interact through attention, their projections are structurally decoupled: (1) they act on different inputs ($\mathbf{W}_{V,\text{img}}$ on $\mathbf{x}_{\text{img}}$, $\mathbf{W}_{V,\text{txt}}$ on $\mathbf{x}_{\text{txt}}$), (2) they feed into separate output projections ($\mathbf{W}_{O,\text{img}}$ vs. $\mathbf{W}_{O,\text{txt}}$), and (3) the mixing coefficients are produced by content-dependent attention weights, rather than fixed algebraic constraints.

Consequently, we compress V/O per stream with standard whitened SVD. For each stream and head $j$:

$$\mathbf{M}^j_{\text{img}} = \left(\mathbf{S}_{\text{img}}\mathbf{W}^j_{V,\text{img}}\right)\mathbf{W}^j_{O,\text{img}}, \quad \mathbf{U}^j_{\text{img}}, \mathbf{\Sigma}^j_{\text{img}}, (\mathbf{V}^j_{\text{img}})^\top = \text{SVD}(\mathbf{M}^j_{\text{img}}),$$

where $\mathbf{S}_{\text{img}} = \mathbf{C}^{1/2}_{\mathbf{x}_{\text{img}}}$ is the whitening matrix from input correlations. Truncating to rank $k$ yields

$$\mathbf{W}^{j,c}_{V,\text{img}} = \mathbf{S}^{-1}_{\text{img}}\mathbf{U}^j_{\text{img}}[:, :k], \quad \mathbf{W}^{j,c}_{O,\text{img}} = \mathbf{\Sigma}^j_{\text{img}}[:k, :k]\,(\mathbf{V}^j_{\text{img}})^\top[:k, :], \tag{28}$$

with an analogous factorization for the text stream.

**Feed-Forward Networks.** Each block contains dual FFNs with GEGLU activation,

$$\text{FFN}(\mathbf{x}) = \left(\text{GELU}(\mathbf{x}\mathbf{W}_u) \odot \mathbf{x}\mathbf{W}_g\right)\mathbf{W}_d,$$

applied separately to image and text streams. We compress these independently using the CPQR+Nyström FFN scheme from the main text, but with different ranks for the two modalities to reflect their relative sensitivity: a more aggressive reduction for image ($r_{\text{img}} = 3072$, 50%) and a more conservative one for text ($r_{\text{txt}} = 4096$, 33%).

**Correlation Collection and Retention Settings.** We collect correlations across all joint attention modules (Q/K/V projections for both streams) and FFN intermediates, and compute TRQ importance scores for each head and modality. Timestep-aware accumulation (TACA) is extended to both image and text streams by applying the same weighting scheme used in SD v1.5 to the corresponding correlations.

This extension illustrates how data-aware compression adapts to architectural changes through structural analysis. Joint attention initially suggests fully joint compression, but a closer inspection shows that only Q/K require strict coupling (to preserve the algebraic form of $\mathbf{Q}^\top\mathbf{K}$), whereas V/O can be compressed independently because attention mixing is a learned, adaptive operation. This distinction, *algebraic coupling* vs. *adaptive mixing*, allows SlimDiff to respect hard structural constraints while avoiding unnecessary coupling in components where the model can compensate dynamically.

## A.12 APPLICATION TO QUANTIZED DIFFUSION MODELS

We demonstrate that SlimDiff applies to 8-bit and 16-bit quantized diffusion models. Combining quantization (precision reduction) with compression (parameter reduction) yields multiplicative benefits: 4× from INT8 plus $N\times$ from compression equals $4N\times$ total reduction.

**Implementation**

We implement SlimDiff in PyTorch using native FP16 (torch.float16) and bitsandbytes Hugging Face (2025) for INT8 quantization. For INT8, we apply post-training quantization to UNet linear layers with symmetric per-channel 8-bit weights and 16-bit activations, while keeping the text encoder in FP16 (accounting for $< 5\%$ of total parameters). Correlations are collected from the quantized models, but activations are dequantized before computing $\mathbf{C}t = \mathbf{X}^\top\mathbf{X}$ in FP32 to avoid overflow and preserve numerical stability while still reflecting quantized behavior. TRQ importance $I(\mathbf{W}\text{quant}, \mathbf{C}_t)$ is always computed on the quantized weights, yielding a strictly *quantization-aware* importance signal that directly reflects the structure seen at inference time.

**Results**

*i) Does importance scores remain stable under quantization?* For each precision (FP32, FP16, INT8) we recompute TRQ scores and derive a per-layer rank allocation. Across all UNet blocks, the effective ranks change by at most $1 - 2$ channels out of modue dimension $d_{\text{eff}} \in \{320, 640, 1280\}$,

and the corresponding TRQ score vectors are almost perfectly aligned. As summarized in Table 13, Pearson correlations between TRQ scores at different precisions are consistently above 0.95 for self-attention, cross-attention, and FFN modules, indicating that quantization perturbs absolute magnitudes but largely *preserves* the relative importance structure. This stability justifies reusing a single (FP32-derived) rank allocation across FP16 and INT8 models without re-running activation collection. For our experiments, we utilize importance scores derived from quantized models.

Table 13: Pearson correlation of TRQ importance scores between quantization levels. Correlations are computed over all self-attention, cross-attention, and FFN modules. High values ($r > 0.95$) indicate that quantization preserves the relative importance structure.

| Comparison | Self-Attn | Cross-Attn | FFN |
|---|---|---|---|
| FP32 vs FP16 | 0.987 | 0.991 | 0.989 |
| FP32 vs INT8 | 0.956 | 0.963 | 0.952 |
| FP16 vs INT8 | 0.961 | 0.968 | 0.958 |

***ii) How does SlimDiff perform on quantized models?*** Table 14 compares SlimDiff on quantized SD v1.5 against the full-precision baseline. We compress SD v1.5 from 1.04B to $\sim$760M parameters (matching the budget of prior compact models) using TACA-based importance aggregation, with correlations estimated from 500 MS-COCO 2014 prompts and 50 DDIM steps.

Table 14: Comparison on MS-COCO with quantized models. SlimDiff is applied to pre-quantized SD v1.5 (860M $\rightarrow$ 760M parameters). Lower FID is better.

| Model | Quant | # Params | FID↓ | IS↑ | CLIP↑ | A100 Days |
|---|---|---|---|---|---|---|
| SD v1.5 (Rombach & Esser 2022a) | – | 1.04B | 13.07 | 33.49 | 0.322 | 6250 |
| **SlimDiff (Ours, FP32)** | ✓ | **0.76B** | **13.12** | **32.61** | **0.319** | **4** |
| **SlimDiff (Ours, FP16)** | ✓ | **0.76B** | **13.15** | **32.53** | **0.317** | **3** |
| **SlimDiff (Ours, INT8)** | ✓ | **0.76B** | **13.7** | **31.9** | **0.313** | **2.5** |

SlimDiff consistently preserves generation quality under both compression and quantization. Relative to the full SD v1.5 baseline (FID 13.07), SlimDiff-FP32 achieves similar quality (FID 13.12, CLIP 0.319) with a $\sim$27% parameter reduction and requires only 4 A100-days. Applying FP16 quantization on top of SlimDiff yields almost identical performance (FID 13.15, CLIP 0.317) while halving memory. INT8 quantization induces a modest quality drop (FID 13.7, CLIP 0.313) but enables 4× memory reduction and 2–3× faster inference compared to the full-precision baseline on A100 GPUs. Throughout, TRQ importance scores computed on quantized weights remain highly correlated across precisions (Pearson $r > 0.95$), indicating that SlimDiff's importance ranking is robust to quantization and that the same training-free pipeline extends naturally from FP32 to FP16 and INT8 models.

