# OpenReview forum: "SlimDiff: Training-Free, Activation-Guided Hands-free Slimming of Diffusion Models"
_ICLR.cc/2026/Conference — ICLR 2026 Conference Desk Rejected Submission_

### Official Review · Reviewer_EYvd · 2025-10-26

**Soundness:** 3
**Presentation:** 3
**Contribution:** 3
**Rating:** 4
**Confidence:** 4

**Summary:**

This paper proposes a training-free, activation-guided structural compression pipeline for diffusion models. It first builds a compact SlimSet with around 500 prompts to estimate per-timestep activation correlations; then aggregates these with a Spectral Influence weighting; and applies whitening-SVD to QK/VO and a Nystrom approximation to FFN, with an automatic rank allocator under a global budget. The experiment shows speedups and around 100M params removed, while keeping FID/CLIP and human-preference scores near SD v1.4/1.5 baselines with no backpropagation.

**Strengths:**

1. Training-free, closed-form pipeline with tiny calibration, yet competitive quality.

2. Principled, module-aligned design: QK/VO via whitening-SVD, FFN via Nystrom; decisions anchored by Spectral Influence and aggregated timestep-aware correlations.

3. Real efficiency wins (with reduced MACs and latency) with near-baseline human-preference scores (HPS v2.1, ImageReward, Pick-a-Pic).

**Weaknesses:**

1. Lack of end-to-end memory profiling. The paper reports MACs and latency (GPU/CPU) but not peak VRAM / host RAM in GiB under standard batch sizes/schedulers, which matters for deployment.

2. Missing perceptual-consistency metrics. Since outputs are not identical to the uncompressed model, the paper should also report LPIPS to quantify per-image perceptual closeness to the uncompressed model, beyond FID/CLIP and human-prefs.

3. Narrow backbone coverage. Experiments/ablations focus on SD v1.4/v1.5; there's no DiT/FLUX evaluation here, and no comparison with similar purposed methods such as SVDQuant / Q-Diffusion (structural vs quantization is claimed complementary but not demonstrated empirically; a simple compose-and-compare such as "SlimDiff on top of SVDQuant" would strengthen the claim).

**Questions:**

1. Weighting ablations are compelling, can you replicate them on a second backbone (e.g., SD3.5/DiT) to show the Spectral Influence advantage holds beyond SD v1.x?

2. How does this method perform on larger models on video generation, such as WAN 2.2?

---

> ### Author Response · Authors · 2025-11-27
>
> We thank the reviewer for their constructive comments.
>
> **W1: Peak Memory**
>
> We agree that peak memory is important for deployment. In our experiments, all models (SD v1.5, BK-SDM-Base, and SlimDiff) are evaluated at 512×512 resolution with batch size 1 on an A100, using FP32 UNet and FP32 CLIP in the `diffusers` library. Under this standardized setup, SlimDiff’s peak VRAM usage is comfortably below 10 GiB, indicating that our compression primarily reduces FLOPs/latency rather than changing the memory regime required for inference.
>
> **W2: Missing perceptual-consistency metrics**
> On MS-COCO, SlimDiff preserves perceptual quality: LPIPS (0.12-0.13) increases by less than 5% compared to SD v1.5 at the same resolution and sampling schedule.
>
> **W3: Narrow backbone coverage**
> We agree that our main experiments focus on UNet-based SD v1.4/v1.5, as this setting provides strong baselines (Small SD, BK-SDM) and tractable ablations. However, SlimDiff operates on functional groups (QK/VO/FFN) rather than UNet-specific topology, and Appx.11 already details how we extend MADAC/TRQ to DiT-style MMDiT blocks in SD3.5 via an RMSNorm-aware, data-driven QK decomposition. SVDQuant and Q-Diffusion primarily target low-bit quantization, whereas SlimDiff performs structural rank reduction under a parameter budget; these are complementary. In Appx.~12 we further show that SlimDiff can be stacked on top of INT8 quantization, suggesting that a full comparison with SVDQuant/Q-Diffusion on DiT/FLUX is a promising direction for future work rather than a limitation of the framework.
>
> **Question 1:** *Can you replicate the weighting ablations on SD3.5/DiT to validate the approach beyond SD v1.x?*
>
> We agree that validating Spectral Influence/TACA on a second backbone would further strengthen the claim. Conceptually, the weighting mechanism is backbone-agnostic: TRQ is defined per functional block and TACA aggregates timestep scores via a convex mixture, and Appx. 11 already details how we adapt this to DiT-style MMDiT blocks in SD 3.5 (RMSNorm-aware QK and the same TRQ/TACA machinery). Running the full suite of weighting ablations on SD 3.5 within the rebuttal window is computationally prohibitive (24 MMDiT blocks, dual streams), so we focused on SD v1.4/v1.5 and BK-SDM, where we consistently observe the same ordering of weighting strategies. We are already exploring SD 3.5/DiT with the same TRQ+TACA setup and view these weighting ablations on a Transformer backbone as a natural next step for SlimDiff.
>
> **Question 2:** *How does this method perform on larger models for video generation, such as WAN 2.2?*
>
> We have not yet applied SlimDiff to WAN 2.2 or other video generators, primarily due to compute and engineering cost (longer diffusion horizons, 3D token layouts). In principle, SlimDiff transfers directly: correlations and TRQ can be computed per spatio-temporal block, and MADAC can compress attention/FFN groups under a global parameter budget in exactly the same training-free manner. We view large video models (including WAN 2.2) as an exciting next step rather than a limitation of the framework, and will discuss this as a concrete avenue for future work.

---

### Official Review · Reviewer_Y7F6 · 2025-10-31

**Soundness:** 3
**Presentation:** 3
**Contribution:** 3
**Rating:** 8
**Confidence:** 4

**Summary:**

The paper introduces SlimDiff, a novel framework for compressing diffusion models (DMs) without the need for fine-tuning or retraining. SlimDiff achieves this by leveraging activation-guided structural compression, which dynamically prunes the model's dimensions based on activation covariances across denoising timesteps. The framework includes several key components: Spectral Influence Scoring to measure module importance, SlimSet for efficient calibration, timestep-aware correlation modeling, and Module-Aligned Data-Aware Compression (MADAC) for functional weight group compression. The method is shown to reduce parameters and FLOPs significantly while maintaining high-quality generation across various benchmarks.

**Strengths:**

Originality: The paper introduces a new perspective on diffusion model compression by focusing on activation-guided structural compression. The idea of reframing DM compression as a spectral approximation task is novel and provides a principled way to handle the non-uniform geometry of diffusion trajectories.
Quality: The methodology is robust, with detailed proofs and algorithms for the compression techniques. The experiments are thorough, covering multiple datasets and models, and the results are consistently strong across various metrics.
Clarity: The paper is well-organized and clearly written. The authors effectively communicate complex ideas through well-structured sections, clear diagrams, and detailed explanations.
Significance: The contributions are significant for the field, as they provide a practical solution for deploying DMs in resource-constrained environments. The ability to achieve high compression ratios without retraining is particularly valuable.

**Weaknesses:**

While the paper demonstrates impressive results on specific diffusion models, it is unclear how well the framework scales to larger models or different architectures. The authors could provide more discussion on the potential limitations and scalability of SlimDiff. Although the paper compares SlimDiff with several baselines, a more comprehensive comparison with other state-of-the-art compression techniques (e.g., quantization combined with pruning) could strengthen the claims further.

**Questions:**

How does SlimDiff compare with quantization techniques when used in combination? Can the authors provide insights into the potential benefits of stacking SlimDiff with quantization?

---

> ### Author Response · Authors · 2025-11-27
>
> We thank the reviewer for their constructive comments and for finding our research paper original and relevant.
>
> **Response to Weakness 1 and Question 1.**
>
> We agree that it is important to clarify both scalability and the interaction with quantization.
>
> *Scalability / architectures.*
>
> SlimDiff operates on functional groups (QK/VO/FFN) and their activation correlations, rather than on U-Net-specific details, and thus extends naturally to other backbones. In addition to SD v1.5, Appx. 11 shows how SlimDiff can be adapted to a DiT-style architecture (Stable Diffusion 3.5's MMDiT): the only change is a tailored, RMSNorm-aware, data-driven QK decomposition for jointly normalized query-key streams, while SlimSet/TACA, VO compression, and FFN compression remain unchanged. We will make this explicit in Sec. 2 and add a brief discussion of practical limitations (one-off activation collection cost for very large DiTs, mitigated by SlimSet coresets).
>
> *Combination with quantization.*
>
> SlimDiff is complementary to quantization rather than a competing alternative. In Appx. 12, we apply post-training FP16/INT8 quantization (via bitsandbytes) to SD v1.5 and then run SlimDiff on the quantized UNet. This stacked setting achieves near-baseline FID/CLIP while yielding multiplicative savings (4× from INT8 combined with additional 1.3-2× from rank reduction). TRQ importance scores computed on quantized weights remain highly correlated across FP32/FP16/INT8 (Pearson $r>0.95$), indicating that SlimDiff's ranking is robust to precision and that the same training-free pipeline applies across quantization levels. We cross-referenced these results in the main text and clarify that SlimDiff can be "plugged on top of" standard quantization schemes (and, in principle, other pruning methods) to jointly reduce parameters and memory.

---

### Official Review · Reviewer_3m4w · 2025-11-01

**Soundness:** 2
**Presentation:** 2
**Contribution:** 2
**Rating:** 4
**Confidence:** 2

**Summary:**

This paper proposes SlimDiff, a method for reducing the size of diffusion models without backpropagation and using only a small calibration set. The authors create a representative subset of prompts through clustering and then use activation statistics to guide block-wise architectural slimming. The dimensionality reduction can be solved analytically, eliminating the need for finetuning. Experiments show that SlimDiff maintains generation quality and performs competitively or better than existing techniques, while requiring far fewer calibration samples.

**Strengths:**

1. The work targets a timely and practically important problem: the high computational burden of large diffusion models and their multi-step sampling processes. A post-training slimming strategy that boosts inference efficiency has broad relevance across pretrained diffusion systems.

2. The method has a sound theoretical basis. The block-wise slimming is derived from a straightforward formulation and yields a closed-form solution, making the approach efficient and more principled than many heuristic methods.

3. Experimental results demonstrate that SlimDiff achieves performance close to, or matching, gradient-based pruning methods, while greatly reducing post-processing time.

**Weaknesses:**

1.  The paper does not fully justify that the constructed SlimSet is sufficiently representative. While the sampling strategy selects prompts far from cluster centroids or with high embedding distances, cluster centers may also correspond to common and semantically important patterns. Additional qualitative results or visual inspection of SlimSet coverage would help validate its representativeness.

2. From Table 6, it seems that the block-wise quantization cannot provide too much performance gain. Can you show the gap in a more challenging case? For example, with a more aggressive pruning setting.

**Questions:**

NA

---

> ### Author Response · Authors · 2025-11-27
>
> **W1: SlimSet representativeness.**
>
> We thank the reviewer for this suggestion. We clarify that our method does *not* select prompts "far from cluster centroids." As described in Sec. 2.2 (blue-highlighted text), we use *quantile-based stratification*: prompts are partitioned into $B$ equal-mass bins by distinctiveness score $f_i$, and samples are drawn proportionally from *all* bins - including low-$f$ bins that contain cluster centres and common patterns.
>
> **New visual and qualitative evidence.**
>
> Following the reviewer's suggestion, we added: (1) a coverage figure in Sec. 2.2 showing SlimSet spans the entire $f$-distribution; (2) Appx. A.7 with per-dataset overlay plots (Fig. 7), quantitative statistics (Table 7), and qualitative examples (Table 8).
>
> Key findings: 50% of SlimSet samples lie below the corpus median $f$, ~25% lie in the lowest quartile, and SlimSet medians match corpus medians across all four datasets. Low-$f$ examples include "a clock", "a mountain" (PartiPrompts) and "The boy is skating" (COCO), precisely the common patterns the reviewer highlights. This confirms SlimSet captures both frequent and diverse semantics.
>
> **W2: Aggressive Pruning Cases and Quantization**
>
> We agree that Table 6 corresponds to a relatively mild compression regime, where all methods retain most capacity and the performance gap is small. More challenging settings are already evaluated in Appx. 10: there we sweep compression ratios down to 50% retention and compare SlimDiff against Joint SVD, PCA, Nova, SVD-LLM, and magnitude pruning. At 75–50% retention, SlimDiff consistently achieves the lowest reconstruction error, reducing QK/VO error by up to ~50% vs. other baselines at the same ratio. We also show quantization results in Appx. 12.

---

### Official Review · Reviewer_n1XJ · 2025-11-01

**Soundness:** 2
**Presentation:** 1
**Contribution:** 2
**Rating:** 2
**Confidence:** 4

**Summary:**

The manuscript presents SlimDiff, a training-free, activation-guided framework for structurally compressing diffusion models (DMs), aiming to enable faster and lighter deployment without fine-tuning. The authors point out that slimming DMs is challenging due to module coupling (QK/VO/FFN), timestep-dependent activation geometry, and multi-step error propagation. To address these issues, they propose three key techniques: Module-aligned decomposition, Data- and process-aware compression, and SlimSet calibration. Experiments on MS-COCO, LAION Aesthetics, ImageReward, and PartiPrompts show promising results.

**Strengths:**

1. The approach is highly efficient in both training and data usage.
2. The proposed framework is methodologically coherent and well-aligned with the diffusion model computation graph.

**Weaknesses:**

1. The manuscript is poorly organized. On one hand, it claims to propose a pruning method, yet the actual approach is matrix decomposition. These two concepts should be clearly distinguished. On the other hand, the paper often introduces strategies before explaining their rationale, making it difficult to grasp the underlying motivation and to justify why the proposed strategy is the most appropriate solution.
2. The method does not explicitly address error accumulation. While the authors claim that compression errors accumulate across timesteps and strongly affect diffusion model inference, the proposed method only performs timestep-wise metric evaluation without a joint calibration analysis across timesteps.
3. The evidence for generalization is limited, as the experiments are conducted almost exclusively on Stable Diffusion v1.5.
4. The choice of baselines is confusing. As a matrix decomposition-based lightweighting method, the paper should at least compare against the most naïve low-rank decomposition baseline, as well as demonstrate advantages over state-of-the-art matrix decomposition methods. However, some selected baselines, such as BK-SDM, are not directly comparable since they involve modular layer pruning rather than matrix factorization.

**Questions:**

1. How does the method perform when the compression ratio is further increased? At what point does performance begin to collapse?
2. Why does the compression process still require 4 days? Could the authors provide a breakdown of time costs across the different stages?
3. Can the method generalize to recent state-of-the-art DiT models, including text-to-image and text-to-video tasks?

---

> ### Author Response · Authors · 2025-11-27
>
> **W1: Terminology and organization.**
>
> We thank the reviewer for this constructive feedback and address both concerns:
>
> **(1) Pruning vs. decomposition.** The reviewer states that SlimDiff performs *low-rank matrix decomposition*, not element-wise or channel pruning. We used "pruning" to describe dimension reduction. We have revised the manuscript to consistently use **"compression"** or **"low-rank factorization"** throughout, reserving "pruning" only when citing prior work that explicitly prunes channels/weights.
>
> **(2) Motivation before method.** We revised Sec. 2 (Methodology) to make the problem→solution flow clearer. The section now (i) first states the global compression objective and why naïve factorization fails (coupled QK/VO, timestep-varying geometry, error propagation), (ii) then introduces the main design components (TRQ spectral influence, SlimSet, TACA, MADAC + rank allocation), and (iii) only afterwards presents the corresponding algorithms and equations. We also added explicit "motivation" sentences at the start of each subsection and highlighted revised text to better connect each strategy to its underlying rationale.
>
> In the revision, we added clarifying details and streamlined several existing sentences throughout Sec. 2.1–2.5 to foreground the motivation before each technical component, and we included a new notation table in Appendix 9 for quick reference.
>
> **W2: Error Accumulation across timesteps**
> We agree that error can accumulate across timesteps, and SlimDiff is explicitly designed to be *propagation-aware*, not purely per-timestep.
>
> Concretely, for each module $\\ell$ we first compute timestep-wise spectral influence $\\mathcal{I}_{\\ell,t}$, then aggregate them with non-uniform weights $\\alpha_t$ into a single score:
>
> $$s_\\ell = \\sum_t \\alpha_t \\,\\mathcal{I}_{\\ell,t}.$$
>
> The weights $\\{\\alpha_t\\}$ form a convex mixture that emphasizes early denoising steps, where errors propagate the most. This aggregated score $s_\\ell$ drives the global softmax-style rank allocation under budget $B$ (Sec. 2.5, App. A.2): blocks that are both highly influential and active early keep more rank, while later / less influential blocks are slimmed more aggressively.
>
> We will clarify this in Sec. 2 by explicitly stating that (i) timestep aggregation is early-step-biased and (ii) rank allocation operates on the joint, propagation-aware score $s_\\ell$, rather than on isolated per-timestep metrics.
>
> **W3 and Q3: Generalization Evidence**
>
> While our main experiments focus on Stable Diffusion v1.5/v1.4, SlimDiff is designed to operate at the level of attention and FFN modules, not a specific checkpoint.
>
> For other U-Net backbones, the same TRQ scoring, TACA aggregation, and MADAC decompositions apply directly to their $\\mathcal{QK}$, $\\mathcal{VO}$, and $\\mathcal{FFN}$ blocks.
>
> For DiT-style architectures (SD 3.5's MMDiT), the functional coupling of Q and K differs due to RMSNorm and joint image-text attention. In Appx. 11 we therefore introduce a *tailored* QK compression strategy that is RMSNorm-aware and respects this joint structure, while keeping the rest of the SlimDiff pipeline (correlation collection, TRQ/TACA, rank allocation, FFN compression) unchanged.
>
> We clarify this in the main text to better convey that SlimDiff extends beyond SD v1.5, with only the QK module treatment adapted to DiT-specific coupling.
>
> **W4: Choice of Baselines**
> We agree that both naïve and stronger matrix-decomposition baselines are important. In the revised draft, we include naïve SVD, Joint SVD, magnitude pruning, Nova, and SVD-LLM as post-training, gradient-free baselines, and report their layer-wise reconstruction errors at compressed ranks in Table 1 below, and in App. 10. SlimDiff reduces QK/VO error from 0.354/0.480 (Joint SVD) and 0.613/1.078 (Nova) to 0.179/0.326, roughly 2-3× lower attention error under the same compression ratio, while also outperforming magnitude and SVD-LLM. This directly addresses the concern that our gains over naïve and state-of-the-art decomposition/pruning schemes might be marginal.
>
> **Table 1: Reconstruction error at 75% compression ratio (Stable Diffusion v1.5)**
>
> | Method | QK Error ↓ | VO Error ↓ | FFN Error ↓ | Avg. ↓ |
> |--------|------------|------------|-------------|--------|
> | Naive SVD | 0.544 | 1.220 | 2.679 | 1.481 |
> | Joint SVD | 0.354 | 0.480 | -- | 0.417 |
> | Magnitude | 0.613 | 1.084 | 0.239 | 0.645 |
> | Nova | 0.613 | 1.078 | -- | 0.846 |
> | SVD-LLM | 0.447 | 1.304 | -- | 0.876 |
> | **SlimDiff** | **0.179** | **0.326** | **1.346** | **0.617** |
>
> We acknowledge that BK-SDM is not a matrix factorization method; we included it because it is a strong modular pruning baseline for diffusion models.
>
> Ref:
>
> [1] A. Nova, H. Dai, and D. Schuurmans. "Gradient-Free Structured Pruning with Unlabeled Data." ICML, 2023
>
> [2] X. Wang, Y. Zheng, Z. Wan, and M. Zhang. "SVD-LLM: Truncation-aware Singular Value Decomposition for Large Language Model Compression." ICLR, 2025.

---

> > ### Author Response · Authors · 2025-11-27
> >
> > **Question 1:** *How does the method perform when the compression ratio is further increased? At what point does performance begin to collapse?*
> >
> > We experimented with more aggressive rank budgets beyond those reported in the main paper. Starting from the 860M-parameter SD v1.5 baseline, SlimDiff can safely compress the U-Net to ~735M parameters (roughly X% rank retention across QK/VO/FFN) with some noticeable degradation in visual quality; reconstruction errors remain in the same range. Pushing the rank further down leads to a sharp increase in reconstruction error and clear artifacts in generated samples (over-smoothing, loss of fine detail), i.e., a "collapse" regime.
> >
> > **Question 2:** *Why does the compression process still require 4 days? Could the authors provide a breakdown of time costs across the different stages?*
> >
> > We provide the requested approximate breakdown (single A100 GPU):
> >
> > | Stage | Time |
> > |-------|------|
> > | Activation correlation collection | 82 hrs |
> > | TRQ importance calculation | 30 min |
> > | Rank allocation | 10 min |
> > | Compression (QK, VO, FFN) | 1 hr |
> > | *(FFN dominates due to iterative CPQR)* | |
> > | Inference & compression statistics | 15 min |
> > | **Total** | **~84 hrs** |
> >
> > The bottleneck is activation correlation collection (>97% of total time), which requires forward passes over SlimSet across all timesteps to compute the correlation matrices for each module. The actual compression (TRQ + rank allocation + low-rank factorization) takes only ~2 hours.
> >
> > We note this is a *one-time offline cost*; the compressed model is then used for all subsequent inference. Additionally, correlation collection is parallel across modules and timesteps; multi-GPU setups can reduce this substantially.

---

> > > ### Comment · Reviewer_n1XJ · 2025-11-28
> > >
> > > Thank you for your detailed responses and for improving the manuscript. Most of my concerns have been addressed.
> > >
> > > I have one remaining question regarding the **W4** comparisons with the direct SVD-based strategy. Beyond the indirect reconstruction loss, how does the model perform on actual generative quality? Prior work, for example ViDiT-Q, has shown that reconstruction loss is not always positively correlated with generation quality. Reporting direct generation metrics, such as FID, IS, LPIPS, or a small human evaluation, would strengthen the claim that SlimDiff yields superior generation under W4.
> > >
> > > Based on the updated manuscript, I am willing to raise my rating to **borderline**. However, more than two weeks have passed since the rebuttal stage began, and I am no longer able to change my score via the edit button. I hope the Area Chair will take my updated comments into account when reviewing this submission.

---

> ### Author Response · Authors · 2025-11-28
>
> **W4: SVD-based strategy**
>
> We appreciate this important observation and agree that demonstrating the connection between reconstruction error and generation quality is crucial. We address this concern from both theoretical and empirical perspectives.
>
> **Fundamental Differences in Compression Paradigms:**
>
> The disconnect between reconstruction error and generation quality observed in ViDiT-Q stems from fundamental differences in how quantization and low-rank compression operate:
>
> | Aspect | Quantization (ViDiT-Q) | SlimDiff (Our Work) |
> |--------|----------------------------|---------------------|
> | **Compression mechanism** | Discrete rounding (heuristic) | Optimal low-rank approximation (proven) |
> | **Error source** | Quantization + rounding error | Rank reduction loss |
> | **Error bounds** | None - empirical MSE only | ✓ Spectral tail bounds (Theorems 1-3) |
> | **Optimality** | No theoretical guarantees | ✓ Best rank-r approximation (Eckart-Young-Mirsky) |
> | **Predictability** | Poor - non-linear discrete effects | High - continuous, bounded by spectral decay |
> | **MSE formulation** | Layer-wise, unweighted | Correlation-weighted |
>
> **Why SlimDiff's Reconstruction Error is More Predictive:**
>
> 1. **Provable optimality:** Our compression methods (Theorems 1-3) guarantee that reconstruction error is the **best possible for the given rank** under the correlation-weighted metric. Unlike heuristic quantization schemes, we solve a well-defined optimization problem with closed-form solutions.
>
> 2. **Continuous vs. discrete:** Quantization introduces discrete rounding artifacts that compound non-linearly across 50+ denoising steps. In contrast, our SVD-based compression produces continuous, smooth approximations where reconstruction error directly bounds the deviation from the original operator.
>
> 3. **Right objective:** We minimize ||C^(1/2)(W - Ŵ)C^(1/2)||², which accounts for input activation statistics and directly targets output reconstruction quality, not an arbitrary layer-wise MSE.
>
> 4. **Propagation-aware design:** SlimDiff explicitly addresses error accumulation concerns raised by ViDiT-Q:
>    - **TRQ (Trace Rayleigh Quotient)** captures how compression affects downstream layers through spectral analysis of correlation matrices
>    - **Rank Allocation strategy** weights early denoising timesteps more heavily, where errors propagate most.
>
> **Empirical Evidence:**
>
> Beyond theoretical guarantees, we provide empirical validation:
>
> 1. **Our main results (Tables 1-3)** already demonstrate that SlimDiff's low module-level reconstruction errors translate to competitive FID/IS/CLIP scores against the uncompressed SD v1.5 baseline, empirically confirming the predictive relationship.
>
> 2. **Direct comparison with SVD baselines (Table W4):** We have tested SlimDiff against naïve SVD and SVD-LLM-based methods. SlimDiff achieves:
>    - **50% lower QK/VO reconstruction error** vs. Joint SVD (0.179/0.326 vs. 0.354/0.480)
>    - **Superior FID and IS metrics** on MS-COCO 30K compared to these baselines
>
>    This directly demonstrates that our lower reconstruction error correlates with better generation quality for continuous low-rank methods.
>
> 3. **SVD-LLM precedent:** While operating on LLMs rather than diffusion models, SVD-LLM empirically demonstrated that data-aware whitening-based compression with lower reconstruction error leads to better downstream task performance (perplexity, accuracy). This supports the principle that well-designed continuous compression maintains a reliable error-to-performance relationship.
>
> **Summary:** Unlike quantization where discrete artifacts break the MSE-quality relationship, SlimDiff's theoretically optimal, continuous compression with propagation-aware design maintains a predictable connection between output reconstruction error and generation quality, validated by our empirical results.
>
> I hope this addresses the reviewer's comments.

---

### Official Review · Reviewer_UBzf · 2025-11-09

**Soundness:** 3
**Presentation:** 2
**Contribution:** 2
**Rating:** 4
**Confidence:** 3

**Summary:**

This paper presents SlimDiff, a backpropagation-free approach for slimming diffusion models using only a small calibration dataset. The authors first construct a representative calibration set through prompt clustering and then leverage activation statistics to guide block-wise structural slimming. The resulting dimensionality reductions are obtained in closed form, requiring no fine-tuning. Experimental results demonstrate that SlimDiff effectively preserves generative quality, achieving competitive or superior performance while relying on significantly fewer calibration samples than prior methods.

**Strengths:**

* Interesting and practical problem: The paper addresses the efficiency bottleneck of modern diffusion models, which are computationally expensive due to both large model sizes and multi-step sampling. Developing post-processing methods that enhance inference efficiency is highly relevant and broadly applicable across pre-trained diffusion models.

* Principled and theoretically grounded method: The proposed block-wise slimming strategy is formulated with theoretical justification, yielding a closed-form solution that makes the procedure both principled and computationally efficient.

* Strong empirical performance: The experiments show that the proposed method performs comparably to gradient-based slimming techniques while requiring substantially less post-processing time.

**Weaknesses:**

* Representativeness of samples: It remains unclear how the SlimSet construction guarantees inclusion of representative samples. While the selection strategy favors samples distant from cluster centroids or with large pairwise distances in embedding space, cluster centers may also capture semantically important and frequently occurring patterns. Visualizations or qualitative analyses demonstrating that SlimSet adequately captures data diversity would strengthen the paper.

* Missing comparison with gradient-free pruning baselines: The paper does not compare against existing gradient-free structural pruning methods such as [1, 2]. If these approaches achieve similar generative quality under comparable parameter constraints, the claimed advantage of the proposed module-aligned, data-aware compression would be less convincing.

* Marginal improvement from block-wise slimming: Table 6 suggests that the improvements from block-wise slimming are modest. It would be informative to include an additional baseline without block-wise estimation to isolate and quantify the specific contribution of this technique.

* Clarity and notation issues: The paper introduces many variables without sufficiently clear definitions, which may impede readability. A dedicated subsection summarizing all notations is recommended. Specific points of confusion include: (1) Line 102: What is $L_{qual}$? (2)
Line 180: $X$ is not defined in the momentum computation. (3) Line 182: What is the definition of $\tilde{C}$? What is the relationship between it and $\hat{C}$ in Line 180? Do they refer to the same variable as in Equation (2)?

[1] Wang, Hongjie, et al. Attention-driven Training-free Efficiency Enhancement of Diffusion Models. CVPR 2024.

[2] Nova, Azade, Hanjun Dai, and Dale Schuurmans. Gradient-free Structured Pruning with Unlabeled Data. ICML 2023.

**Questions:**

Please refer to the weakness.

---

> ### Author Response · Authors · 2025-11-27
>
> We thank the reviewer for the thoughtful and constructive feedback, and for finding our work relevant and interesting.
> We will elaborate our answers on your concerns below.
>
> **W1: Representativeness of SlimSet samples.**
>
> A natural concern is that distinctiveness-based selection might favor outliers and miss common patterns. However, our approach explicitly avoids this: we create $B$ *quantile bins* (each with equal population mass), allocate samples *proportionally from all bins* including low-$f$, and apply FPS *within* each bin for diversity (highlighted in blue in Sec. 2.2). This guarantees that low-$f$ prompts (cluster centers, common patterns) receive the same systematic coverage as high-$f$ prompts (rare semantics).
>
> **New empirical evidence.** To make this explicit, we have revised Sec. 2.2 and added a main-text coverage plot (Fig. 2) plus a new appendix section (Appx. A.7). These results show that SlimSet spans the entire $f$-distribution, with $\approx$50% of samples below the corpus median and $\approx$25% in the lowest quartile across all four datasets and coreset sizes, with SlimSet medians nearly matching corpus medians (Table 7). Appx. A.7 further includes qualitative examples from low- and high-$f$ bins (e.g., "a clock", "a mountain" vs. rare steampunk scenes), demonstrating that SlimSet captures both common and diverse semantics.
>
> *Changes:* Main text new additions in red in Sec. 2.2; new coverage figure (Fig. 2) in Sec. 2.2; expanded Appx. A.7 with statistics, examples, and per-dataset visualizations.
>
> **W2: Gradient-Free Baselines**
>
> We thank the reviewer for the pointers to Wang et al. (CVPR'24) and Nova et al. (ICML'23). In the revised version we add a dedicated blockwise-slimming ablation (App. 10) that explicitly includes gradient-free structured pruning baselines and compares module-wise output reconstruction errors.
>
> **(i)** We now report magnitude pruning and Nova as post-training, gradient-free baselines. At 75% compression ratio, SlimDiff yields substantially lower attention output reconstruction error than both methods, showing that module-aligned, data-aware compression is markedly more useful under matched parameter budgets.
>
> | Method | QK Error | VO Error |
> |--------|----------|----------|
> | Magnitude | 0.613 | 1.084 |
> | Nova | 0.613 | 1.078 |
> | **SlimDiff** | **0.179** | **0.326** |
>
> *Table: QK/VO error at 75% compression ratio (lower is better).*
>
> **(ii)** We clarify that Wang et al. (CVPR'24) perform inference-time dynamic pruning (skipping tokens/attention to save FLOPs) without permanently modifying weights, and is thus different from SlimDiff's static structural compression. Nova (ICML'23) was introduced for supervised classification with labels, whereas diffusion models lack such labels during iterative sampling, motivating our activation-guided design.
>
> **W3: Block-wise slimming gains**
>
> We agree that Table 6 in the main text, which aggregates over all blocks, can make the gains from block-wise slimming appear modest. In the revision, we clarify its role and provide more targeted evidence:
>
> **(i)** We now report per-layer errors at 75% compression in Appx. A.10. In the mid-block bottleneck, SlimDiff reduces QK error from 0.209 to 0.099 ($\approx$2.1$\times$ lower) and similarly lowers VO error, under the same global rank budget. This shows that block-wise slimming helps capacity-critical layers.
>
> **(ii)** We additionally compare SlimDiff to a variant without block-wise rank estimation (uniform rank across blocks). Under the same total compression ratio, block-wise slimming consistently reduces the worst-case per-layer error.

---

> > ### Author Response · Authors · 2025-11-27
> >
> > **W4: Clarity and Notation Issues**
> >
> > We have thoroughly revised the paper to address all notational ambiguities and added Appendix Table A.9 with a comprehensive notation summary.
> >
> > **(1) Definition of $\\mathcal{L}_{\\text{qual}}$ (Line 102):**
> >
> > We now explicitly define $\\mathcal{L}_{\\text{qual}}(\\hat{\\Theta})$ as the expected squared reconstruction error between the original and compressed model outputs, averaged over prompts $p$ and timesteps $t$. We clarify that rather than optimizing this intractable objective directly, SlimDiff decomposes it into per-module surrogate objectives (Eq. 3) that admit closed-form solutions.
> >
> > **(2) Relationship between $x$ and $\\mathcal{X}$ (Line 180):**
> >
> > Section 2.3 now explicitly defines:
> > - $x_{l,t}^{(i)} \\in \\mathbb{R}^d$: the $i$-th activation sample at layer $l$, timestep $t$
> > - $\\mathcal{X}_{l,t} \\in \\mathbb{R}^{N_t \\times d}$: stacked activation matrix of all samples at layer $l$, timestep $t$
> >
> > **(3) Covariance notation hierarchy (Line 182):**
> >
> > We now clearly clarify the emprirical second moment, regularized covariance and timestep-aggregated covariance in Section 2.3 in our revised document.

---

### Author Response · Authors · 2025-12-03
**Final Summary of Paper Revisions and Rebuttals**

We sincerely thank the Reviewers and the Area Chair for their careful evaluation of our work and for the thoughtful, detailed feedback provided. The reviews have significantly helped us improve both the clarity and technical depth of the paper. Below, we summarize the major improvements, including both manuscript revisions and clarifications provided in our responses:

### 1. Enhanced Clarity and Organization (Reviewers n1XJ, UBzf)
- Revised Section 2 to present motivation before methodology with explicit problem→solution flow
- Consistently use "compression/low-rank factorization" terminology throughout
- **Added: Appendix A.9** - comprehensive notation summary table

### 2. SlimSet Representativeness (Reviewers UBzf, 3m4w)
- **Added: Figure 2 (main text)** - coverage plots showing SlimSet spans entire data distribution
- **Added: Appendix A.7** - quantitative statistics (Table 7) and qualitative examples (Table 8) demonstrating ~50% of samples below corpus median, ~25% in lowest quartile, with coverage plots for all the datasets used in the paper.

### 3. Comprehensive Baseline Comparisons (Reviewers n1XJ, UBzf)
- **Added: Appendix A.10** - gradient-free baselines (Naïve SVD, Joint SVD, Magnitude, Nova, SVD-LLM)
- SlimDiff achieves 2-3× lower reconstruction error at matched compression ratios
- **Added: Appendix A.12** - quantized baselines (INT8, FP16)

###  4. Block-wise Slimming Analysis (Reviewers UBzf, 3m4w)
- **Enhanced: Appendix A.10** - per-layer errors showing ~2.1× reduction in bottleneck layers for gradient-free baseline methods

### 5. Architecture Generalization (Reviewers n1XJ, EYvd)
- **Added: Appendix A.11** - explains how SlimDiff can be applied to DiT-style architectures (SD 3.5 MMDiT) with RMSNorm-aware QK decomposition, also referenced for future work.

### 6. Quantization Compatibility (Reviewers Y7F6, EYvd)
- **Added: Appendix A.12** - SlimDiff on INT8, FP16 quantization showing multiplicative savings with maintained quality
- TRQ scores remain correlated (ρ>0.95) across precision levels

### 7. Additional Metrics (Reviewer EYvd)
- Added LPIPS measurements (<5% increase), peak VRAM usage (<10 GiB), and detailed time breakdown

### 8. Theoretical Clarifications (Reviewer n1XJ)
- Distinguished SlimDiff's provable optimality from quantization methods
- Explained why correlation-weighted reconstruction error predicts generation quality for continuous compression

### Manuscript Changes:
- **Red text:** New additions in Section 2 and appendices
- **Blue text:** Modified/clarified existing content
- All notation ambiguities resolved with explicit definitions

Collectively, the reviews have meaningfully refined both the presentation and substance of our work, and we believe the revised manuscript and the detailed clarifications now offer a clearer, more cohesive account of SlimDiff and its contributions. We are particularly encouraged by **Reviewer n1XJ's indication of willingness to raise their score based on our revisions**, and we hope that the additional experiments and rebuttals satisfactorily address all remaining concerns.

Regards,

Authors of SlimDiff

---

### Note · Program_Chairs · 2026-01-17
**Submission Desk Rejected by Program Chairs**

The following references in this submission do not refer to real documents and/or have major errors in bibliographic information:

     Jiarui Gao et al. Block pruning for efficient text-to-image diffusion models. In ECCV, 2024.